# Ridge Rider: Finding Diverse Solutions by Following Eigenvectors of the Hessian

**Jack Parker-Holder**[*]
University of Oxford

**Luke Metz**
Google Research, Brain Team

**Cinjon Resnick**
NYU

**Hengyuan Hu**
FAIR

**Adam Lerer**
FAIR

**Alistair Letcher**

**Alex Peysakhovich**
FAIR

**Aldo Pacchiano**
BAIR

**Jakob Foerster**[*]
FAIR

## Abstract

Over the last decade, a single algorithm has changed many facets of our lives - Stochastic Gradient Descent (SGD). In the era of ever decreasing loss functions, SGD and its various offspring have become the go-to optimization tool in machine learning and are a key component of the success of deep neural networks (DNNs). While SGD is guaranteed to converge to a local optimum (under loose assumptions), in some cases it may matter *which* local optimum is found, and this is often context-dependent. Examples frequently arise in machine learning, from shape-versus-texture-features to ensemble methods and zero-shot coordination. In these settings, there are desired solutions which SGD on 'standard' loss functions will not find, since it instead converges to the 'easy' solutions. In this paper, we present a different approach. Rather than following the gradient, which corresponds to a locally greedy direction, we instead follow the eigenvectors of the Hessian, which we call "ridges". By iteratively following and branching amongst the ridges, we effectively span the loss surface to find qualitatively different solutions. We show both theoretically and experimentally that our method, called *Ridge Rider* (RR), offers a promising direction for a variety of challenging problems.

## 1 Introduction

Deep Neural Networks (DNNs) are extremely popular in many applications of machine learning ranging from vision [23, 49] to reinforcement learning [47]. Optimizing them is a non-convex problem and so the use of gradient methods (e.g. stochastic gradient descent, SGD) leads to finding local minima. While recent evidence suggests [9] that in supervised problems these local minima obtain loss values close to the global minimum of the loss, there are a number of problem settings where optima with the same value can have very different properties. For example, in Reinforcement Learning (RL), two very different policies might obtain the same reward on a given task, but one of them might be more robust to perturbations. Similarily, it is known that in supervised settings some minima generalize far better than others [28, 24]. Thus, being able to find a specific *type* or class of minimum is an important problem.

At this point it is natural to ask what the benefit of finding diverse solutions is? Why not optimize the property we care about directly? The answer is *Goodhart's law*: "When a measure becomes a target, it ceases to be a good measure." [48]. Generalization and zero-shot coordination are two examples of these type of objectives, whose very definition prohibits direct optimization.

---

[*]Equal contribution. Correspondence to jackph@robots.ox.ac.uk , jnf@fb.com

To provide a specific example, in computer vision it has been shown that solutions which use shape features are known to generalize much better than those relying on textures [18]. However, they are also more difficult to find [16]. In reinforcement learning (RL), recent work focuses on constructing agents that can coordinate with humans in the cooperative card game Hanabi [4]. Agents trained with self-play find easy to learn, but highly arbitrary, strategies which are impossible to play with for a novel partner (including human). To avoid these undesirable minima previous methods need access to the symmetries of the problem to make them inaccessible during training. The resulting agents can then coordinate with novel partners, including humans [26]. Importantly, in both of these two cases, standard SGD-based methods do not find these 'good' minima easily and problem-specific hand tuning is required by designers to prevent SGD from converging to 'bad' minima.

Our primary contribution is to take a step towards addressing such issues in a general way that is applicable across modalities. One might imagine a plausible approach to finding different minima of the loss landscape would be to initialize gradient descent near a saddle point of the loss in multiple replicates, and hope that it descends the loss surface in different directions of negative curvature. Unfortunately, from any position near a saddle point, gradient descent will always curve towards the direction of most negative curvature (see Appendix D.1), and there may be many symmetric directions of high curvature.

Instead, we start at a saddle point and *force* different replicates to follow distinct, orthogonal directions of negative curvature by iteratively following each of the eigenvectors of the Hessian until we can no longer reduce the loss, at which point we repeat the branching process. Repeating this process hopefully leads the replicates to minima corresponding to distinct convex subspaces of the parameter space, essentially converting an optimization problem into search [30]. We refer to our procedure as Ridge Rider (RR). RR is less 'greedy' than standard SGD methods, with Empirical Risk Minimization (ERM, [51]), and thus can be used in a variety of situations to find diverse minima. This greediness in SGD stems from it following the path with highest expected local reduction in the loss. Consequently, some minima, which might actually represent the solutions we seek, are never found.

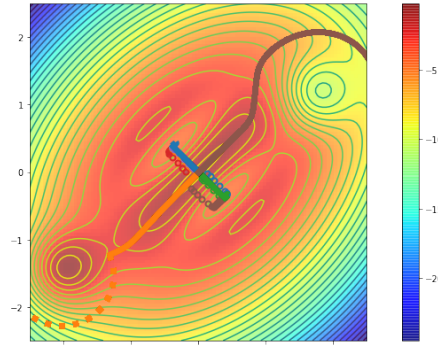

Figure 1: Comparison of gradient descent (GD, hollow circles) and RR (RR, solid circles) on a two-dimensional loss surface. GD starting near the origin only finds the two local minima whose basin have large gradient near the origin. RR starts at the maximum and explores along four paths based on the two eigenvectors of $\mathcal{L}$. Two paths (blue and green) find the local minima while the other two explore the lower-curvature ridge and find global minima. Following the eigenvectors leads RR around a local minimum (brown), while causing it to halt at a local maximum (orange) where either a second ride (dotted) or GD may find a minimum.

In the next section we introduce notation and formalize the problem motivation. In Section 3 we introduce RR, both as an exact method, and as an approximate scalable algorithm. In both cases we underpin our approach with theoretical guarantees. Finally, we present extensions of RR which are able to solve a variety of challenging machine learning problems, such as finding diverse solutions reinforcement learning, learning successful strategies for zero-shot coordination and generalizing to out of distribution data in supervised learning. We test RR in each of these settings in Section 4. Our results suggest a conceptual connection between these previously unrelated problems.

## 2   Background

Throughout this paper we assume a smooth, *i.e.*, infinitely differentiable, loss function $\mathcal{L}_\theta = \mathbb{E}_{\mathbf{x}_i} L_\theta(\mathbf{x}_i)$, where $\theta \in \mathbb{R}^n = \Theta$ are typically the weights of a DNN. In the supervised setting, $\mathbf{x}_i = \{x_i, y_i\}$ is an input and label for a training data point, while $L_\theta(\mathbf{x}_i)$ could be the cross-entropy between the true label $y_i$ and the prediction $f_\theta(x_i)$.

We use $\nabla_\theta \mathcal{L}$ for the gradient and $\mathcal{H}$ for the Hessian, ie. $\mathcal{H} = \nabla_\theta^2 \mathcal{L}$. The eigenvalues (EVals) and eigenvectors (EVecs) of $\mathcal{H}$ are $\lambda_i$ and $e_i$ respectively. The computation of the full Hessian is prohibitively expensive for all but the smallest models, so we assume that an automatic differentiation library is available from which vector-Hessian products, $\mathcal{H}v$, can be computed efficiently [43]: $\mathcal{H}v = \nabla_\theta\big((\nabla_\theta \mathcal{L})v\big)$.

**Symmetry and Equivalence**   Real world problems commonly contain symmetries and invariances. For example, in coordination games, the payout is unchanged if all players jointly update their strategy to another equilibrium with the same payout. We formalize this as follows: A symmetry, $\phi$, of the loss function is a bijection on the parameter space such that $\mathcal{L}_\theta = \mathcal{L}_{\phi(\theta)}$, for all $\theta \in \Theta$.

If there are $N$ non-overlapping sets of $m$ parameters each, $k_i = \{\theta_{k_i^1}, ..\theta_{k_i^m}\}, i \in \{1, ...N\}$ and the loss function is invariant under all permutations of these $N$ sets, we call these sets 'N-equivalent'. In a card-game which is invariant to color/suit, this corresponds to permuting both the color-dependent part of the input layer and the output layer of the DNN simultaneously for all players.

**Zero-Shot Coordination**   The goal in Zero-Shot Coordination is to coordinate with a stranger on a fully cooperative task during a single try. Both the problem setting and the task are common knowledge. A zero-shot coordination scheme (or learning rule) should find a policy that obtains high average reward when paired with the distribution of policies obtained independently but under the same decision scheme. All parties can agree on a scheme beforehand, but then have to obtain the policy independently when exposed to the environment. A key challenge is that problems often have symmetries, which can generate a large set of equivalent but mutually incompatible policies.

We assume we have a fully observed MDP with states $s_t \in \mathcal{S}$ and agents $\pi_{i\in[1,N]}$, each of whom chooses actions $a_t^i \in \mathcal{A}$ at each step. The game is cooperative with agents sharing the reward $r_t$ conditioned on the joint action and state, and the goal is to maximize expected discounted return $J = \mathbb{E}_\tau \sum_t \gamma^t r_t$, where $\gamma$ is the discount factor and $\tau$ is the trajectory.

**Out of Distribution Generalization (OOD)**   We assume a multi-environment setting where our goal is to find parameters $\theta$ that perform well on all $N$ environments given a set $m < N$ of training environments $\Xi = \{\xi_0, \xi_1, \ldots, \xi_m\}$. The loss function $\ell$ and the domain and ranges are fixed across environments. Each environment $\xi$ has an associated dataset $D_\xi$ and data distribution $\mathcal{D}_\xi$. Together with our global risk function, $l$, these induce a per environment loss function,

$$\mathcal{L}_\xi(\theta) = \mathbb{E}_{\mathbf{x}_i \sim \mathcal{D}_\xi} \ell(\mathbf{x}_i).$$

Empirical Risk Minimization (ERM) ignores the environment and minimizes the average loss across all training examples,

$$\mathcal{L}_{ERM}(\theta) = \mathbb{E}_{\mathbf{x}_i \sim \cup_{\xi \in \Xi} \mathcal{D}_\xi} \ell_\xi(\mathbf{x}_i).$$

ERM can fail when the test distribution differs from the train distribution.

# 3   Method

We describe the intuition behind RR and present both exact and approximate (scalable) algorithms. We also introduce extensions to zero-shot coordination in multi-agent settings and out of distribution generalization.

The goal of RR is as follows. Given a smooth loss function, $\mathcal{L}(\theta)$, discover qualitatively different local minima, $\theta^*$, while grouping together those that are equivalent (according to the definition provided in Section 2) or, alternatively, only exploring one of each of these equivalent policies.

While symmetries are problem specific, in general *equivalent parameter sets* (see Section 2) lead to repeated EVals. If a given loss function in $\mathbb{R}^n$ has $N$-equivalent parameter sets ($N > 2$) of size $m$ and a given $\theta$ is invariant under all the associated permutations, then the Hessian at $\theta$ has at most $n - m(N - 2)$ distinct eigenvalues (proof in Appendix D.6). Thus, rather than having to explore up to $n$ orthogonal directions, in some settings it is sufficient to explore one member of each of the groups of distinct EVals to obtain different non-symmetry-equivalent solutions. Further, the EVecs can be ordered by the corresponding EVal, providing a numbering scheme for the classes of solution.

To ensure that there is at least one negative EVal and that all negative curvature directions are locally loss-reducing, we start at or near a strict saddle point $\theta^{\text{MIS}}$. For example, in supervised settings, we can accomplish this by initializing the network near zero. Since this starting point combines small gradients and invariance, we refer to it as the Maximally Invariant Saddle (MIS). Formally,

$$\theta^{\text{MIS}} = \arg\min_\theta |\nabla_\theta J(\theta)|, \text{ s.t. } \phi(\theta) = \theta, \ \forall\phi,$$

for all symmetry maps $\phi$ as defined in Section 2.

In tabular RL problems the MIS can be obtained by optimizing the following objective (Proof in Appendix D.7):

$$\theta^{\text{MIS}} = \arg\min_{\theta} |\nabla_{\theta} J(\theta)| - \lambda H(\pi_{\theta}(\mathbf{a})), \lambda > 0$$

From $\theta^{\text{MIS}}$, RR proceeds as follows: We branch to create replicas, which are each updated in the direction of a different EVec of the Hessian (which we refer to as 'ridges'). While this is locally loss reducing, a single step step typically does not solve a problem. Therefore, at all future timesteps, rather than choosing a new EVec, each replicate follows the updated version of its original ridge until a break-condition is met, at which point the branching process is repeated. For any ridge this procedure is repeated until a locally convex region without any negative curvature is found, at which point gradient descent could be performed to find a local optimum or saddle. In the latter case, we can in principle apply RR again starting at the saddle (although we did not find this to be necessary in our setting). We note that RR effectively turns a continuous optimization problem, over $\theta$, into a discrete search process, *i.e.*, which ridge to explore in what order.

During RR we keep track of a *fingerprint*, $\Psi$, containing the indices of the EVecs chosen at each of the preceding branching points. $\Psi$ uniquely describes $\theta^{\Psi}$ up to repeated EVals. In Algorithm 1 we show pseudo code for RR, the functions it takes as input are described in the next paragraphs.

UpdateRidge computes the updated version of $e_i$ at the new parameters. The EVec $e_i$ is a continuous function of $\mathcal{H}$ ([27], pg 110-111).[1] This allows us to 'follow' $e_i(\theta_t^{\Psi})$ (our 'ridge') for a number of steps even though it is changing. In the exact version of RR, we recompute the spectrum of the Hessian after every parameter update, find the EVec with greatest overlap, and then step along this updated direction. While this updated EVec might no longer correspond to the $i$-th EVal, we maintain the subscript $i$ to index the ridge. The dependency of $e_i(\theta)$ and $\lambda_i(\theta)$ on $\theta$ is entirely implicit: $\mathcal{H}(\theta)e_i(\theta) = \lambda_i(\theta)e_i(\theta)$, $|e_i| = 1$.

EndRide is a heuristic that determines how long we follow a given ridge for. For example, this can consider whether the curvature is still negative, the loss is decreasing and other factors.

GetRidges determines which ridges we explore from a given branching point and in what order. Note that from a saddle, one can explore in *opposite* directions along any negative EVec. Optionally, we select the $N$ most negative EVals.

ChooseFromArchive provides the search-order over all possible paths. For example, we can use breadth first search (BFS), depth first search (DFS) or random search. In BFS, the archive is a FIFO queue, while in DFS it is a LIFO queue. Other orderings and heuristics can be used to make the search more efficient, such as ranking the archive by the current loss achieved.

---

**Algorithm 1** Ridge Rider

1: **Input:** $\theta^{\text{MIS}}$, $\alpha$, ChooseFromArchive, UpdateRidge, EndRide, GetRidges
2: $A = [\{\theta^{\Psi=[i]}, e_i, \lambda_i\}$ for $i, e_i, \lambda_i \in \text{GetRidges}(\theta^{\text{MIS}})]$      // Initialize Archive of Solutions
3: **while** $|A| > 0$ **do**
4:      $\{e_i, \theta_0^{\Psi}, \lambda_i\}, A = \text{ChooseFromArchive}(A)$      // Select a ridge from the archive
5:      **while** True **do**
6:          $\theta_t^{\Psi} = \theta_{t-1}^{\Psi} - \alpha e_i$      // Step along the Ridge with learning rate $\alpha$
7:          $e_i, \lambda_i = \text{UpdateRidge}(\theta_t^{\Psi}, e_i, \lambda_i)$      // Get updated Ridge
8:          **if** $\text{EndRide}(\theta_t^{\Psi}, e_i, \lambda_i)$ **then**
9:              break      // Check Break Condition
10:          **end if**
11:      **end while**
12:      $A = A \cup \left[\{\theta^{\Psi.\text{append}(i)}, e_i, \lambda_i\}$ for $i, e_i, \lambda_i \in \text{GetRidges}(\theta^{\Psi})\right]$      // Add new Ridges
13: **end while**

---

It can be shown that, under mild assumptions, RR maintains a descent direction: At $\theta$, its EVals are $\lambda_1(\theta) \geq \lambda_2(\theta) \cdots \geq \lambda_d(\theta)$ with EVecs $e_1(\theta), \cdots, e_d(\theta)$. We denote the eigengaps as $\Delta_{i-1} := \lambda_{i-1}(\theta) - \lambda_i(\theta)$ and $\Delta_i := \lambda_i(\theta) - \lambda_{i+1}(\theta) = \Delta_i$, with the convention $\lambda_0 = \infty$.

**Theorem 1.** *Let $L : \Theta \to \mathbb{R}$ have $\beta-$smooth Hessian (i.e. $\|\mathcal{H}(\theta)\|_{op} \leq \beta$ for all $\theta$), let $\alpha$ be the step size. If $\theta$ satisfies: $\langle \nabla L(\theta), e_i(\theta) \rangle \geq \|\nabla L(\theta)\|\gamma$ for some $\gamma \in (0,1)$, and $\alpha \leq \frac{\min(\Delta_i, \Delta_{i-1})\gamma^2}{16\beta}$ then after two steps of RR:*

$$L(\theta'') \leq L(\theta) - \gamma\alpha\|\nabla L(\theta)\|$$

*Where $\theta' = \theta - \alpha e_i(\theta)$ and $\theta'' = \theta' - \alpha e_i(\theta')$.*

In words, as long as the correlation between the gradient and the eigenvector RR follows remains large, the slow change in eigenvector curvature will guarantee RR remains on a descent direction. Further, starting from any saddle, after $T$-steps of following $e_i(\theta_t)$, the gradient is $\nabla\mathcal{L}(\theta_T) = \alpha \sum_t \lambda_i(\theta_t)e_i(\theta_t) + \mathcal{O}(\alpha^2)$. Therefore, $\langle \nabla\mathcal{L}(\theta), e_i(\theta_T) \rangle = \alpha \sum_t \lambda_i(\theta_t)\langle e_i(\theta_t), e_i(\theta_T) \rangle + \mathcal{O}(\alpha^2)$ Thus, assuming $\alpha$ is small, a sufficient condition for reducing the loss at every step is that $\lambda_i(\theta_t) < 0, \forall t$ and the $e_i(\theta_t)$ have positive overlap, $\langle e_i(\theta_t), e_i(\theta_{t'}) \rangle > 0, \forall t, t'$. Proofs are in Appendix D.4.

**Approximate RR:** In exact RR above, we assumed that we can compute the Hessian and also obtain all EVecs and EVals. To scale RR to large DNNs, we make two modifications. First, in $\mathrm{GetRidges}$ we use the power method (or Lanczos method [19]) to obtain approximate versions of the N most negative $\lambda_i$ and corresponding $e_i$. Second, in $\mathrm{UpdateRidge}$ we use gradient descent after each parameter update $\theta^\Psi \to \theta^\Psi - \alpha e_i$ to yield a new $e_i, \lambda_i$ pair that minimizes the following loss:

$$L(e_i, \lambda_i; \theta) = |(1/\lambda_i)\mathcal{H}(\theta)e_i/|e_i| - e_i/|e_i||^2$$

We warm-start with the 1st-order approximation to $\lambda(\theta)$, where $\theta', \lambda', e_i'$ are the previous values:

$$\lambda_i(\theta) \approx \lambda_i' + e_i'\Delta\mathcal{H}e_i' = \lambda_i' + e_i'(\mathcal{H}(\theta) - \mathcal{H}(\theta'))e_i'$$

Since these terms only rely on Hessian-Vector-products, they can be calculated efficiently for large scale DNNs in any modern auto-diff library, *e.g.* Pytorch [41], Tensorflow [1] or Jax [6]. See Algorithm 2 in the Appendix (Sec. C) for pseudocode.

We say $e_0 = e_i(\theta)$ and $e_t$ is the $t-$th EVal in the algorithm's execution. This algorithm has the following convergence guarantees, *ie.*:

**Theorem 2.** *If $L$ is $\beta-$smooth, $\alpha_e = \min(1/4, \Delta_i, \Delta_{i-1})$, and $\|\theta - \theta'\| \leq \frac{\min(1/4, \Delta_i, \Delta_{i-1})}{\beta}$ then*

$$|\langle e_t, e_i(\theta')\rangle| \geq 1 - \left(1 - \frac{\min(1/4, \Delta_i, \Delta_{i-1})}{4}\right)^t$$

This result characterizes an exponentially fast convergence for the approximate RR optimizer. If the eigenvectors are well separated, UpdateRidge will converge faster. The proof is in Appendix D.3.

**RR for Zero-Shot Coordination in Multi-Agent Settings:** RR provides a natural decision scheme for this setting – decide in advance that each agent will explore the top $F$ fingerprints. For each fingerprint, $\Psi$, run $N$ independent replicates $\pi$ of the RR procedure and compute the average cross-play score among the $\pi$ for each $\Psi$. At test time, deploy a $\pi$ corresponding to a fingerprint with the highest score. Cross-play between two policies, $\pi^a$ and $\pi^b$, is the expected return, $J(\pi_1^a, \pi_2^b)$, when agent one plays their policy of $\pi^a$ with the policy for agent two from $\pi^b$.

This solution scheme relies on the fact that the ordering of unique EVals is consistent across different runs. Therefore, fingerprints corresponding to polices upon which agents can reliably coordinate will produce mutually compatible polices across different runs. Fingerprints corresponding to arbitrary symmetry breaking will be affected by inconsistent EVal ordering since EVals among equivalent directions are equal.

Consequently, there are two key insights that makes this process succeed without having to know the symmetries. The first is that the MIS initial policy is invariant with respect to the symmetries of the task, and the second is that equivalent parameter sets lead to repeated EVals.

**Extending RR for Out of Distribution Generalization:** Consider the following coordination game. Two players are each given access to a non-overlapping set of training environments, with the goal of learning consistent features. While both players can agree beforehand on a training scheme and a network initialization, they cannot communicate after they have been given their respective datasets. This coordination problem resembles OOD generalization in supervised learning.

RR could be adapted to this task by finding solutions which are reproducible across all datasets. One necessary condition is that the EVal and EVec being followed is consistent across training

environments $\xi \in \Xi$ to which each player has access:

$$\mathcal{H}_\xi e = \lambda e, \ \forall \xi \in \Xi, \ \lambda < 0$$

where $\mathcal{H}_\xi$ is the Hessian of the loss evaluated on environment $\xi$, *i.e.*, $\nabla_\theta^2 \mathcal{L}_\xi$. Unfortunately, such $e, \lambda$ do not typically exist since there are no consistent features present in the raw input. To address this, we extend RR by splitting the parameter space into $\Theta_f$ and $\Theta_r$. The former embeds inputs as **f**eatures for the latter, creating an abstract feature space representation in which we can run RR.

For simplicity, we consider only two training environments with, respectively, Hessians $\mathcal{H}_r^1$ and $\mathcal{H}_r^2$. The $R$ indicates that we are only computing the Hessian in the subspace $\Theta_r$. Since we are not aware of an efficient and differentiable method for finding common EVecs, we parameterize a differentiable loss function to optimize for an approximate common EVec, $e_r$, of the Hessians of all training environments in $\Theta_r$. This loss function forces high correlation between $H_r e_r$ and $e_r$ for both environments, encourages negative curvature, prevents the loss from increasing, and penalizes differences in the EVals between the two training environments:

$$\mathcal{L}_1(\theta_f, e_r | \theta_r) = \sum_{i \in 1,2} \left( -\beta_1 \mathrm{C}(\mathcal{H}_r^i e_r, e_r) - \beta_2 e_r \mathcal{H}_r^i e_r + \beta_3 L_{\xi_i}(\theta_f | \theta_r) \right) + \frac{|e_r(\mathcal{H}_r^1 - \mathcal{H}_r^2)e_r|}{|e_r|^2}.$$

Here C is the correlation (a normalized inner product) and the "$(\cdot|\theta_r)$" notation indicates that $\theta_r$ is not updated when minimizing this loss, which can be done via stop_gradient. All $\beta_i$ are hyperparameters. The minimum of $\mathcal{L}_1(\theta_f, e_r | \theta_r)$ is a consistent, negative EVal/EVec pair with low loss.

For robustness, we in parallel train $\theta_f$ to make the Hessian in $\theta_r$ consistent across the training environments in other directions. We do this by sampling random unit vectors, $u_r$ from $\Theta_r$ and comparing the inner products taken with the Hessians from each environment. The loss, as follows, has a global optimum at 0 when $\mathcal{H}^1 = \mathcal{H}^2$:

$$\mathcal{L}_2(\theta_f | \theta_r) = \mathbb{E}_{u_r \sim \Theta_r} \beta_4 \frac{|\mathrm{C}(\mathcal{H}_r^1 u_r, u_r)^2 - \mathrm{C}(\mathcal{H}_r^2 u_r, u_r)^2|}{\mathrm{C}(\mathcal{H}_r^1 u_r, u_r)^2 + \beta_5 \mathrm{C}(\mathcal{H}_r^1 u_r, u_r)^2} + \frac{|u_r(\mathcal{H}_r^1 - \mathcal{H}_r^2)u_r|}{|u_r \mathcal{H}_r^1 u_r| + |u_r \mathcal{H}_r^2 u_r|}.$$

RR now proceeds by starting with randomly initialized $e_r$, $\theta_f$, and $\theta_r$. Then we iteratively update $e_r$ and $\theta_f$ by running n-steps of SGD on $\mathcal{L}_1 + \mathcal{L}_2$. We then run a step of RR with the found approximate EVec $e_r$ and repeat. Pseudo-code is provided in the Appendix (Sec. C.3).

**Goodhart's Law, Overfitting and Diverse Solutions** As mentioned in Section 1, RR directly relates to Goodhart's law, which states that any measure of progress fails to be useful the moment we start optimizing for it. So while it is entirely legitimate to use a validation set to estimate the generalization error for a DNN trained via SGD *after the fact*, the moment we directly optimize this performance via SGD it seizes to be informative.

In contrast, RR allows for a two step optimization process: We first produce a finite set of diverse solutions using only the training set and then use the validation data to chose the best one from these. Importantly, at this point we can use any generalization bound for finite hypothesis classes to bound our error [35]. For efficiency improvements we can also use the validation performance to locally guide the search process, which makes it unnecessary to actually compute all possible solutions.

Clearly, rather than using RR we could try to produce a finite set of solutions by running SGD many times over. However, typically this would produce the same type of solution and thus fail to find those solutions that generalize to the validation set.

# 4 Experiments

We evaluate RR in the following settings: exploration in RL, zero-shot coordination, and supervised learning on both MNIST and the more challenging Colored MNIST problem [3]. In the following section we introduce each of the settings and present results in turn. Full details for each setting are given in the Appendix (Sec. B).

**RR for Diversity in Reinforcement Learning** To test whether we can find diverse solutions in RL, we use a toy binary tree environment with a tabular policy (see Fig. 2). The agent begins at $s_1$, selects actions $a \in \{\mathrm{left}, \mathrm{right}\}$, receiving reward $r \in \{-1, 10\}$ upon reaching a terminal node. For

the loss function, we compute the expectation of a policy as the sum of the rewards of each terminal node, weighted by the cumulative probability of reaching that node. The maximum reward is 10.

We first use the exact version of RR and find $\theta^{\mathrm{MIS}}$, by maximizing entropy. For the $\mathrm{ChooseFromArchive}$ precedure, we use BFS. In this case we have access to exact gradients so can cheaply re-compute the Hessian and EVecs. As such, when we call $\mathrm{UpdateRidge}$ we adapt the learning rate $\alpha$ online to take the largest possible step while preserving the ridge, similar to Backtracking Line Search [39]. We begin with a large $\alpha$, take a step, recompute the EVecs of the Hessian and the maximum overlap $\delta$. We then and sequentially halve $\alpha$ until we find a ridge satisfying the $\delta_{\mathrm{break}}$ criteria (or $\alpha$ gets too small). In addition, we use the following criteria for $\mathrm{EndRide}$: (1) If the dot product between $e_i$ and $e'_i$ is less than $\delta_{\mathrm{break}}$ (2) If the policy stops improving.

We run RR with a maximum budget of $T = 10^5$ iterations similarity $\delta_{\mathrm{break}} = 0.95$, and take only the top $N = 6$ in $\mathrm{GetRidges}$. As baselines, we use gradient descent (GD) with random initialization, GD starting from the MIS, and random norm-one vectors starting from the MIS. All baselines are run for the same number of timesteps as used by RR for that tree. For each depth $d \in \{4, 6, 8, 10\}$ we randomly generate 20 trees and record the percentage of positive solutions found.

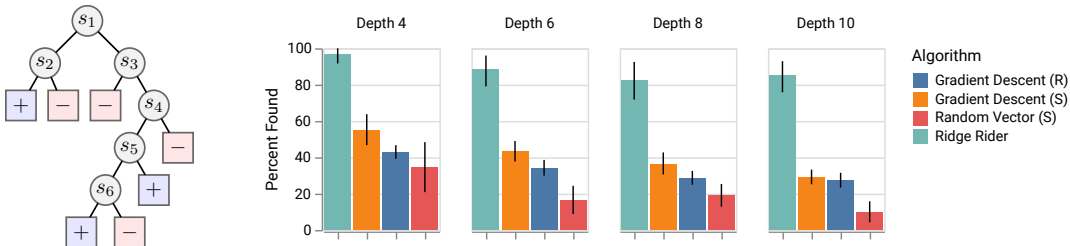

Figure 2: **Left**: a tree with six decision nodes and seven terminal nodes, four of which produce negative rewards (red) and three of which produce positive rewards (blue). **Right**: The percentage of solutions found per algorithm, collated by tree depth. R and S represent starting from a random position or from a saddle, respectively. Trees at each depth are randomly generated 20 times to produce error estimates shown.

On the right hand side of Fig 2, we see that RR outperforms all three baselines. While RR often finds over 90% of the solutions, GD finds at most 50% in each setting. Importantly, following random EVecs performs poorly, indicating the importance of using the EVecs to explore the parameter space. To run this experiment, see the notebook at https://bit.ly/2XvEmZy.

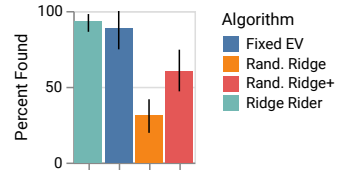

Figure 3: Tree depth 12, ten seeds.

Next we include two additional baselines: (1) following EVecs, but not updating them (*Fixed-EV*). (2) following random unit vectors with *positive ascent direction* (*Rand-Ridge+*), and compare vs. RR. We ran these with a fixed budget, for a tree of depth 12. We used the same hyperparameters for RR and the ablations. As we see in Fig. 3, Fixed-EVs obtains competitive performance. This clearly illustrates the importance of following EVs rather than random directions.

Finally, we open the door to using RR in deep RL by computing Hessians using samples, leveraging more accurate higher order gradients produced by the DiCE objective [15]. Once again, RR is able to find more diverse solutions than SGD (see Fig 8).

**RR for Supervised Learning** We applied approximate RR to MNIST with a 2-layer MLP containing 128 dimensions in the hidden layer. As we see on the left hand side of Fig 4, we found that we can achieve respectable performance of approximately 98% test and train accuracy. Interestingly, updating $e$ to follow the changing eigenvector is crucial. A simple ablation which sets $\mathrm{LR}_e$ to 0 fails to train beyond 90% on MNIST, even after a large hyper parameter sweep (see the right side of Fig 4). We also tested other ablations. As in RL, we consider using random directions. Even when we force the random vectors to be ascent directions (Rand.Ridge +), the accuracy does not exceed 30%. In fact, the outperformance from RL is more pronounced in MNIST, which is intuitive since random search is known to scale poorly to high dimensional problems. We expect this effect to be even more pronounced as Approximate RR is applied to harder and higher dimensional tasks in the future.

With fixed hyperparameters and initialization, the order in which the digit classes are learned changes according to the fingerprint. This is seen in Fig 5. The ridges with low indices (i.e. EVecs with very negative curvature) correspond to learning '0' and '1' initially, the intermediate ridges correspond

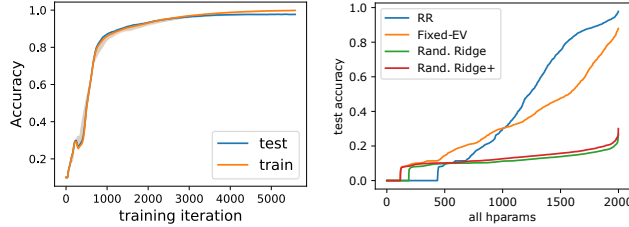

Figure 4: **Left**: Test and training accuracy on MNIST. Our hyperparameters for this experiment were: $S = 236$, $\alpha = 0.00264$, $\mathrm{LR}_x = 0.000510$, $\mathrm{LR}_\lambda = 4.34e^{-6}$, batch size $= 2236$. **Right**: We compare a hyperparameter sweep for approximate RR on MNIST with a simple ablation: Rather than updating the ridge (EVec), we set $\mathrm{LR}_e$ to 0, i.e. keep following the original direction of the EVec. Shown are the runs that resulted in over $> 60\%$ final test accuracy out of a hyper parameter sweep over 2000 random trials. We note that updating the ridge is absolutely crucial of obtaining high performance on MNIST - simply following the fixed eigenvectors with an otherwise unchanged RR algorithm never exceeds the performance of a linear classifier.

to learning '2' and '3' first, and the ridges at the upper end of the spectrum we sampled (ie. $> 30$) correspond to learning features for the digit "8".

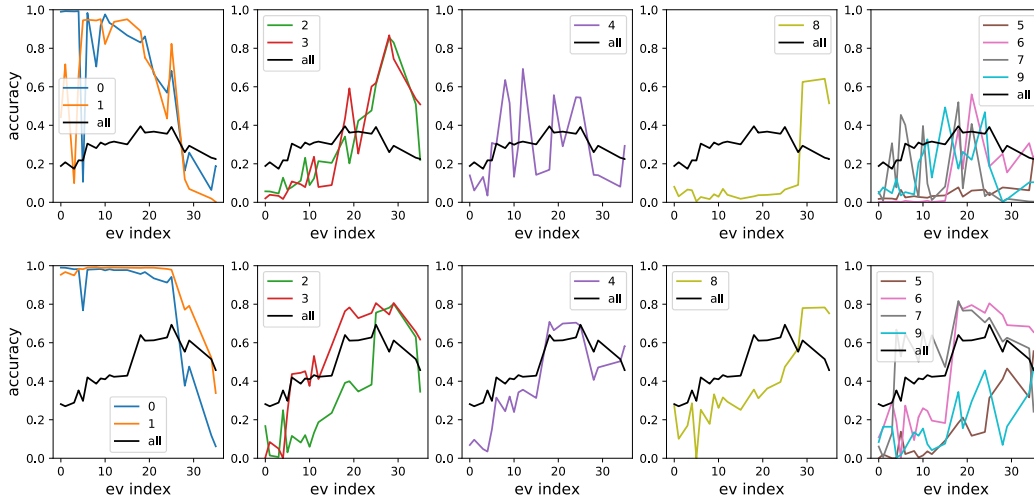

Figure 5: Class accuracy for different digits as a function of the index of the first ridge ($\psi[0]$), i.e the ranking of the EVal corresponding to the first EVec we follow. **Top**: Early in training – average between 200 and 600 steps. **Bottom**: Later in training, averaged between $4000 : 5000$ steps. The architecture is the same MLP as in Figure 4, but the hyperparameters are: $S = 1$, $\alpha = 0.000232$, $\mathrm{LR}_x = 3.20e^{-6}$, $LR_\lambda = 0.00055$, batch size $= 2824$.

**RR for Zero-Shot Coordination** We test RR as described in Sec. 3 on multi-agent learning using the lever coordination game from [26]. The goal is to maximize the expected reward $J$ when playing a matrix game with a stranger. On each turn, the two players individually choose from one of ten levers. They get zero reward if they selected different levers and they get the payoff associated with the chosen lever if their choices matched. Importantly, not all levers have the same payoff. In the original version, nine of the ten levers payed $1$ and one paid $.9$. In our more difficult version, seven of the ten levers pay $1$, two 'partial coordination' levers pay $0.8$, and one lever uniquely pays $0.6$. In self-play, the optimal choice is to pick one of the seven high-paying levers. However, since there are seven equivalent options, this will fail in zero-shot coordination. Instead, the optimal choice is to pick a lever which obtains the highest expected payoff when paired with any of the equivalent policies. Like in the RL setting, we use a BFS version of exact RR and the MIS is found by optimizing for high entropy and low gradient.

We see in Fig 6 that RR is able to find each solution type: the self-play choices that do poorly when playing with others (ridges 1, 3-5, 7-9), the 'partial coordination' that yield lower reward overall (ridges 2 and 6), and the ideal coordinated strategy (ridge 0). Note that based on this protocol

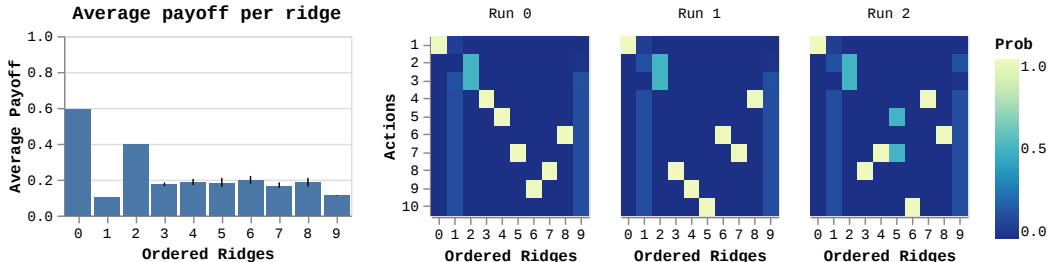

Figure 6: Zero-Shot Coordination: On the left, we see the average payoff per ridge over 25 runs, repeated five times to yield error estimates. As expected, the highest payoff is 0.6 and it occurs when both agents find the symmetry breaking solution, even though that solution yields the lowest payoff in self-play. On the right, we see the results of three randomly chosen runs where each square is a probability that *one* of the two agents select that action. We verified that the agents in each run and each ridge agree on the greedy action.

the agent will chose the 0.6 lever, achieving perfect zero-shot coordination. To run the zero-shot coordination experiment, see the notebook at https://bit.ly/308j2uQ.

**RR for Out of Distribution Generalization**   We test our extension of RR from Sec 3 on OOD generalization using Colored MNIST [3]. Following Sec 2, for each of $\xi_{1,2}$ in $\Xi$, as well as test environment $\xi_3$, $x_i \in D_{\xi_k}$ are drawn from disjoint subsets of MNIST [31] s.t. $|D_{\xi_{1,2}}| = 25000$ and $|D_{\xi_3}| = 10000$. Further, each has an environment specific $p_{\xi_k}$ which informs $\mathcal{D}_{\xi_k}$ as follows: For each $x_i \in D_{\xi_k}$, first assign a preliminary binary label $\tilde{y}_i$ to $x_i$ based on the digit – $\tilde{y}_i = 0$ for $x_i \in [0, 4]$ and $\tilde{y}_i = 1$ for $x_i \in [5, 9]$. The actual label $y_i$ is $\tilde{y}_i$ but flipped with probability .25. Then, sample color id $z_i$ by flipping $y_i$ with probability $p_{\xi_k}$, where $p_{\xi_1} = 0.2$, $p_{\xi_2} = 0.1$, and $p_{\xi_3} = 0.9$. Finally, color the image red if $z_i = 1$ or green if $z_i = 0$. Practically, we first optimize $\mathcal{L}_1 + \mathcal{L}_2$ to find the MIS, resampling the DNN when optimization fails to obtain a low loss.

Chance is 50%. The optimal score is 75% on train *and* test. Fitting a neural network with ERM and SGD yields around 87% on train and 18% on test because it only finds the spurious color correlation. Methods which instead seek the more causal digit explanation achieve about 66% − 69%[3, 2, 29]. As shown in Table 1, our results over 30 runs achieve a high after nine steps along the ridge of 65.5%±1.68 on train and 58.4%±2.41 on test. Our results are clearly both above chance and in line with models that

| Method | Train Acc | Test Acc |
|---|---|---|
| **RR** | $65.5 \pm 1.68$ | $58.4 \pm 2.41$ |
| ERM | $87.4 \pm 1.70$ | $17.8 \pm 1.33$ |
| IRM | $69.7 \pm .710$ | $65.7 \pm 1.42$ |
| Chance | 50 | 50 |
| Optimal | 75 | 75 |

Table 1: **Colored MNIST**: Accuracies on a 95% confidence interval. RR is in line with causal solutions.

find the causal explanation rather than the spurious correlative one. To run the out of distribution generalization experiment, see the notebook at https://bit.ly/3gWeFsH. See Fig 9 in the Appendix for additional results.

## 5   Discussion

We have introduced RR, a novel method for finding specific types of solutions, which shows promising results in a diverse set of problems. In some ways, this paper itself can be thought of as the result of running the breadth-first version of RR - a set of early explorations into different directions of high curvature, which one day will hopefully lead to SotA results, novel insights and solutions to real world problems. However, there is clearly a long way to go. Scaling RR to more difficult problems will require a way to deal with the noise and stochasticity of these settings. It will also require more efficient ways to compute eigenvalues and eigenvectors far from the extreme points of the spectrum, as well as better understanding of how to follow them robustly. Finally, RR hints at conceptual connections between generalization in supervised learning and zero-shot coordination, which we are just beginning to understand. Clearly, symmetries and invariances in the Hessian play a crucial, but under-explored, role in this connection.

## Acknowledgements and Disclosure of Funding

We'd like to thank Brendan Shillingford, Martin Arjovsky, Niladri Chatterji, Ishaan Gulrajani and C. Daniel Freeman for providing feedback on the manuscript. The authors did not receive funding which could create a conflict of interest.

## Broader Impact

We believe our method is the first to propose following the eigenvectors of the Hessian to optimize in the parameter space to train neural networks. This provides a stark contrast to SGD as commonly used across a broad spectrum of applications. Most specifically, it allows us to seek a variety of solutions more easily. Given how strong DNNs are as function approximators, algorithms that enable more structured exploration of the range of solutions are more likely to find those that are semantically aligned with what humans care about.

In our view, the most significant advantage of that is the possibility that we could discover the minima that are not 'shortcut solutions' [16] like texture but rather generalizable solutions like shape [18]. The texture and shape biases are just one of many problematic solution tradeoffs that we are trying to address. This also holds for non-causal/causal solutions (the non-causal or correlative solution patterns are much easier to find) as well as concerns around learned biases that we see in applied areas across machine learning. All of these could in principle be partially addressed by our method.

Furthermore, while SGD has been optimized over decades and is extremely effective, there is no guarantee that RR will ever become a competitive optimizer. However, maybe this is simply an instance of the 'no-free-lunch' theorem [53] - we cannot expect to find diverse solutions in science unless we are willing to take a risk by not following the locally greedy path. Still, we are committed to making this journey as resource-and time efficient as possible, making our code available and testing the method on toy environments are important measures in this direction.

## Footnotes

[1] We rely on the fact that the EVecs are continuous functions of $\theta$; this follows from the fact that $\mathcal{L}(\theta)$ is continuous in $\theta$, and that $\mathcal{H}(\mathcal{L})$ is a continuous function of $\mathcal{L}$.

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
