[Supplementary Material]

# Appendix: Ridge Rider

## A    Related Work

The common approach to finding a specific type of solution to a DNN optimization problem is to modify SGD via a range of algorithmic approaches to initialization, update rules, learning rate, and so forth [8, 28, 37, 38]. By contrast, RR does not follow the gradient at all - instead of pursuing the locally *greedy* direction that SGD seeks, we follow an eigenvector of the Hessian which allows us to directly control for curvature. Wang* et al. [52] also adjusts the solutions found when training neural networks by taking advantage of the Hessian to stay on 'ridges'. However, they focus on minimax optimization such as in the GAN [20] setting. This difference in motivation leads them to an algorithm that looks like Gradient Descent Ascent but with a correction term using the Hessian that keeps the minimax problem from veering into problematic areas with respect to convergence. This is different from our approach, which moves in the direction of the Hessian's eigenvectors instead of gradients. Eigen Vector Descent (EVD, [46]) proposes to update neural network parameters in the direction of an eigenvector of the Hessian, by doing line search on each individual eigenvector at each step and taking the best value. This can be seen as a myopic version of our method, which greedily updates a single policy. By contrast, we do deep exploration of each eigenvector and maintain a set of candidate solutions.

Finally, there are also other optimization approaches that do not rely on gradients such as BFGS and Quasi-Newton. They efficiently optimize towards the nearest solution, sometimes by using the Hessian, but do not do what we are proposing of using the eigenvectors as directions to follow. Rather, they use variants of line search with constraints that include the Hessian.

Motivation for RR stems from a number of different directions. One is in games and specifically the zero-shot coordination setting. Hu et al. [26] presents an algorithm that achieves strong results on the Hanabi benchmark [4] including human-AI coordination, but require problem specific knowledge of the game's symmetries. These symmetries correspond to the arbitrary labelings of the particular state/action space that leave trajectories unchanged up to those labelings. In contrast, RR discovers these symmetries automatically by exploiting the connection between equivalent solutions and repeated Eigenvalues, as we demonstrate empirically in Sec 4.

Another motivating direction is avoiding the 'shortcut' solutions often found by DNNs. These are often subpar for downstream tasks [16, 17, 5, 11, 22]. Geirhos et al. [18] tries to address this concern by adjusting the dataset to be less amenable to the texture shortcut and more reliant on shape. In the causal paradigm, Arjovsky et al. [3] assumes access to differing environments and then adjusts the SGD loss landscape to account for all environments simultaneously via a gradient norm penalty. We differ by looking for structured solutions by following the curvature of the Hessian.

In RL, many approaches make use of augmented loss functions to aid exploration, which are subsequently optimized with SGD. These include having a term boosting diversity with respect to other agents [32, 33, 10, 25, 40, 36, 44] or measuring 'surprise' in the environment [34, 13, 50, 42, 45, 7]. Rather than shifting or augmenting the loss landscape in this way, we gain diversity through structured exploration with eigenvectors of the Hessian. Finally, maximum entropy objectives are also a popular way to boost exploration in RL [21, 12]. However, this is typically combined with SGD rather than used as an initialization for RR as we propose.

# B  Additional Experimental Results

In this section we include some addiitonal results from the diversity in RL experiments.

**Intuition and Ablations** To illustrate the structured exploration of RR, we include a visualization of the optimization path from the MIS. On the left hand side of Fig 7 we show the path along two ridges for the tree shown earlier alongside the main results (Fig 2). Areas of high reward are in dark blue. The ridges (green and light blue) both correspond to distinct positive solutions. The policies have six dimensions, but we project them into two dimensions by forming a basis with the two ridges. Observe that the two ridges are initially orthogonal, following the $(x, y)$ axes. Conversely, we also show two runs of GD from the same initialization, each of which find the same solution and are indistinguishable in parameter space.

Figure 7: **Left**: Example optimization paths. Dark is higher reward. All methods start at the same saddle. The deterministic GDs follow the same trajectory while the RRs follow different paths. **Right**: The percentage of solutions found per algorithm, collated by tree depth. Trees at each depth are randomly generated 20 times to produce error estimates shown.

We also must consider whether we need to adapt $\alpha$, the size of the step along the ridge. In larger scale settings this may not be possible, so we now consider the two following ablations:

1. *Fixed + UpdateRidge*: Here we continue to update the ridge, but use a fixed $\alpha$.
2. *Fixed + FixedRidge*: We not only keep a fixed $\alpha$ but also do not update the ridge. Thus, we take repeated small steps along the original ridge until we meet the $\mathrm{EndRide}$ condition.

In both cases we use $\alpha = 0.1$, the same value used for the Gradient Descent and Random Vector baselines in Fig. 2. As we see in Fig. 7, as we impose greater rigidity, the performance declines. However, the results for the fixed ridge are still stronger than any of the baselines from Fig. 2, showing the power of the original ridges in finding diverse solutions.

**Beyond Exact RL** Next we open the door to scaling our method to the deep RL paradigm. For on-policy methods we can compute $\mathcal{H}$ accurately via the DiCE operator [15, 14]. As a first demonstration of this, we consider the same tree problem shown in Fig. 2, with the same tabular/linear policy. However, instead of computing the loss as an expectation with full knowledge of the tree, we sample trajectories by taking stochastic actions of the policy in the environment. We use the same maximum budget of $T = 10^5$, but using 100 samples to compute the loss function. For RR we alternate between updating the policy by stepping along the ridge and updating the value function (with SGD), while for the baseline we optimize a joint objective. Both RR and the baseline use the Loaded DiCE [14] loss function for the policy. For RR, we use the adaptive $\alpha$ from the exact setting. The results are shown in Fig. 8.

For tree depths of four and six, RR significantly outperforms the vanilla actor critic baseline, with gains also coming on larger trees. It is important to caveat that these results remain only indicative of larger scale performance, given the toy nature of the problem. However, it does show it is possible to follow ridges when the Hessian is computed with samples, thus, we believe it demonstrates the *potential* to scale RR to the problems we ultimately wish to solve. G

Figure 8: The percentage of solutions found per algorithm, collated by tree depth. Trees at each depth are randomly generated 10 times to produce error estimates shown.

# C  Implementation Details

In the following subsections, we provide implementation details along with pseudo code for the approximate version of RR (used in MNIST experiments) and the extensions to Zero-Shot coordination and Colored MNIST, respectively.

## C.1  Approximate RR

---
**Algorithm 2** Scalable RR

---
1: **Input:** $N$ number of ridges; $R$ maximum index of ridge considered; $T$ max iterations per ridge; $S$ inner steps; $\text{LR}_{e/\lambda}$, learning rate for EVec and EVal
2: **Initialize:** Sample neural network $\theta \sim N(0, \epsilon)$ small value, i.e. near saddle.
3: // Find a ridge.
4: $e, \lambda, r = \text{GetRidges}(\theta)$: Sample random integer $r \in [1, R]$ and mini-batch $(\boldsymbol{x}, \boldsymbol{y})$,
5: then use the power method to find the $r$-th most negative Eval $\lambda$ and its EVec $e$.
6: Optionally: Compute gradient $\boldsymbol{g} = \frac{\partial}{\partial \theta} L(\boldsymbol{y}, f_\theta(\boldsymbol{x}))$ and set ridge $e = \text{sign}(e \cdot \boldsymbol{g})e$.
7: archive $= [\{\theta^{[r]}, e, \lambda\}]$
8: **for** 1 to N **do**
9:     $\theta^\Psi, \lambda, e = $ archive.pop(0) // ChooseFromArchive is trivial - there is only one entry
10:     // Follow the ridge.
11:     **while** True **do**
12:         $\theta_{old} = \theta^\Psi$
13:         $\theta^\Psi = \theta - lr_\theta\, e$
14:         $\lambda = \lambda + e^T(\mathcal{H}(\theta^\Psi) - \mathcal{H}(\theta_{old}))e$
15:         UpdateRidge // Update eigenvalue and eigenvector.
16:         Sample a mini-batch $\boldsymbol{x}, \boldsymbol{y}$.
17:         // Starting from $\lambda, e$ use gradient descent to obtain updated values:
18:         **for** 1 to S **do**
19:             $\mathcal{L}(e, \lambda | \theta^\Psi) = (\|(1/\lambda)\mathcal{H}(\theta^\Psi)e/\|e\|_2 - e/\|e\|_2\|_2^2)$
20:             $\lambda = \lambda - \text{LR}_\lambda \frac{\partial}{\partial \lambda}\mathcal{L}(e, \lambda | \theta^\Psi)$
21:             $e = e - \text{LR}_e \frac{\partial}{\partial e}\mathcal{L}(e, \lambda | \theta^\Psi)$
22:         **end for**
23:         **if** $\text{EndRide}(\theta^\Psi, e, \lambda)$ **then**
24:             **break**
25:         **end if**
26:     **end while**
27:     $e, \lambda, r = \text{GetRidges}(\theta^\Psi)$
28:     archive $=[\{\theta^{\Psi.\text{append}(r)}, e, \lambda\}]$
29: **end for**
30: **return** $\theta^\Psi$

---

## C.2  Multi-Agent Zero-Shot Coordination

---
**Algorithm 3** Zero-Shot Coordination

---
1: **Input:** $N$ independent runs; List $\boldsymbol{\Psi} = \{\Psi_1, \dots \Psi_N\}$ fingerprints considered; $T$ max iterations per ridge; Learning rate $\alpha$.
2: **Initialize:** Array solutions = [][], best_score = -1.
3: **for** $i \in 1$ to N **do**
4:     **Initialize:** Policy $\theta \sim N(0, \epsilon)$.
    // Get the Maximally Invariant Saddle (min gradient norm, max entropy).
5:     MIS $= \text{GetMIS}(\theta)$
6:     $s = \text{RidgeRiding}(\text{MIS}, \boldsymbol{\Psi}, \text{T}, \alpha)$ //Run exact version of RR
    // For each $\Psi$, select $\theta$ with highest reward in self-play. Note, multiple $\theta$ per $\Psi$ correspond to $\pm$ EVec directions.
7:     **for** $k \in 1$ to $\text{len}(\boldsymbol{\Psi})$ **do**
8:         solutions$[i][k] = \text{argmax}_{\theta \text{ s.t. } \theta^\Psi == \boldsymbol{\Psi}[k]} J(\theta)$
9:     **end for**
10: **end for**
11: **for** $k \in 1$ to $\text{len}(\boldsymbol{\Psi})$ **do**
12:     **Initialize:** average_score = 0 // Average cross-play score for fingerprint $\Psi$
13:     **for** $i \in \{1, \dots, N\}$ **do**
14:         **for** $j \in \{1, \dots, N\}$ **do**
15:             average_score $+= J(\text{solutions}[i][k]_1, \text{solutions}[j][k]_2)/N^2$
16:         **end for**
17:     **end for**
18:     **if** average_score > best_score **then**
19:         best_score = average_score
20:         $\theta^* = $ solutions[0][k]
21:     **end if**
22: **end for**
23: **return** $\theta^*$

---

## C.3 Colored MNIST

---

**Algorithm 4** Colored MNIST

---

1: **Input:** Training Environments $\xi_1, \xi_2$; Inner steps S; Ridge steps N; Pre-Training Steps H; Loss hyperparameters $\beta$; Number of featurizer weights $f$; Learning rate for featurizer and EVec $LR_{f/x}$; Learning rate for RR $\alpha$; Learning rates for finding MIS in pre-training $\gamma$.

2: **Initialize:** Neural network $\theta \sim N(0, \epsilon)^n$ to small random values, ie. near saddle; candidate common EVec $e_r \sim N(0, \epsilon)^r$ where $r = n - f$.
   // Split weights into featurizer and RR space.

3: $\theta_f = \theta[0:f]$

4: $\theta_r = \theta[f:n]$

5: // Find the Maximally Invariant Saddle and initial common EVec.

6: **for** 1 to H **do**

7: $\quad \mathcal{L}_1(\theta_f, e_r | \theta_r) = \sum_{i \in 1,2} \left( -\beta_1 C(\mathcal{H}_r^i e_r, e_r) - \beta_2 e_r \mathcal{H}_r^i e_r + \beta_3 L_{\xi_i}(\theta_f | \theta_r) \right) + \frac{|e_r(\mathcal{H}_r^1 - \mathcal{H}_r^2)e_r|}{|e_r|^2}$

8: $\quad \mathcal{L}_2(\theta_f | \theta_r) = \mathbb{E}_{u_r \sim \Theta_r} \beta_4 \frac{|C(\mathcal{H}_r^1 u_r, u_r)^2 - C(\mathcal{H}_r^2 u_r, u_r)^2|}{C(\mathcal{H}_r^1 u_r, u_r)^2 + \beta_5 C(\mathcal{H}_r^1 u_r, u_r)^2} + \frac{|u_r(\mathcal{H}_r^1 - \mathcal{H}_r^2)u_r|}{|u_r \mathcal{H}_r^1 u_r| + |u_r \mathcal{H}_r^2 u_r|}$

9: $\quad$ // $\mathcal{H}_r^\xi = \nabla_\theta^2 \mathcal{L}_\xi$ and $C$ is the correlation (normalized dot product)

10: $\quad \theta_f = \theta_f - \gamma_0 \frac{\partial}{\partial \theta_f}(L_1 + L_2)$

11: $\quad e_r = e_r - \gamma_1 \frac{\partial}{\partial e_r}(L_1)$

12: **end for**

13: **for** 1 to N **do**

14: $\quad$ // Follow the ridge.

15: $\quad \theta_r = \theta_r - \alpha x$
   // Update EVec and featurizer.

16: $\quad$ **for** 1 to S **do**

17: $\quad\quad \theta_f = \theta_f - LR_f \frac{\partial}{\partial \theta_f}(L_1 + L_2)$

18: $\quad\quad e_r = e_r - LR_x \frac{\partial}{\partial e_r}(L_1)$

19: $\quad$ **end for**

20: **end for**

---

Figure 9: Curves showing the training and test accuracy of IRM (Invariant Risk Minimization), ERM (Empirical Risk Minimization), and RR on Colored MNIST. Note that the bottom x-axis is steps for ERM/IRM and the top x-axis is steps for RR.

## D  Theoretical Results

### D.1  Behavior of gradient descent near a saddle point

We will illustrate how gradient descent dynamics near a saddle point moves towards the most negative eigenvector of the Hessian via two different derivations. These are not novel results but provided as an illustration of a well known fact.

First, let $\theta_0$ be a saddle point of $\mathcal{L}(\theta)$, and consider $T$ steps of gradient descent $\theta_1, ..., \theta_T$. Let $\mathcal{H}(\theta_0)$ be the Hessian of $\mathcal{L}$ at $\theta_0$. We will use the first-order Taylor expansion, $\nabla_\theta \mathcal{L}(\theta_t) = H(\theta_t - \theta_0) + o(\epsilon^2)$ ignoring the error term to approximate the gradient close to $\theta_0$.

We can decompose $\theta_t - \theta_0$ into the basis of eigenvectors of $\mathcal{H}(\theta_0)$: $\theta_t - \theta_0 = \sum_i a_{i,t} e_i(\theta_0)$ where $\{e_i(\theta_0)\}$ are the eigenvectors of $\mathcal{H}(\theta_0)$. After one step of gradient descent with learning rate $\alpha$,

$$\theta_{t+1} = \theta_0 + \sum_i a_{i,t} e_i(\theta_0) - \alpha \sum_i \lambda_i a_{i,t} e_i(\theta_0) \tag{1}$$

$$= x_0 + \sum_i (1 - \alpha \lambda_i(\theta_0)) a_{i,t} e_i(\theta_0) \tag{2}$$

i.e. $a_{i,t+1} = (1 - \alpha \lambda_i(\theta_0)) a_{i,t}$

It follows by simple induction that if $T$ isn't too large so the displacement $\theta_T - \theta_0$ is still small, $a_{i,T} = (1 - \alpha \lambda_i)^T$. In other words, the component of $\theta_1 - \theta_0$ corresponding to more negative eigenvalues of $\mathcal{H}(\theta_0)$ will be amplified relative to less negative eigenvalues by a ratio that grows exponentially in $T$.

In the limit of small step sizes, we can also consider the differential limit of approximate (up to first order terms as defined above) gradient descent dynamics

$$\frac{d\theta}{dt} = -\alpha \mathcal{H}(\theta_0)\theta,$$

assuming the saddle is at $\theta_0 = 0$ wlog. The solution to this system of equations is $\theta(t) = \theta(0)\exp(-\alpha \mathcal{H}(\theta_0)t)$. If we write the eigendecomposition of $\mathcal{H}(\theta_0)$, $\mathcal{H}(\theta_0) = Q\Lambda Q^{-1}$, then $\exp(-\mathcal{H}(\theta_0)) = -Q\exp(\Lambda)Q^{-1}$. So if $\theta(0) = \sum_i a_i e_i(\theta_0)$, then $\theta(t) = \sum_i e^{-\alpha \lambda_i(\theta_0)t} a_i e_i(\theta_0)$.

### D.2  Structural properties of the eigenvalues and eigenvectors of Smooth functions

**Definition 1.** *We say a function $f : \mathbb{R}^d \to \mathbb{R}$ is $\beta-$smooth if for all pairs of points $\theta, \theta' \in \mathbb{R}^d$:*
$$|\nabla_\theta^2 f(\theta) - \nabla_\theta^2 f(\theta')| \leq \beta \|\theta - \theta'\|$$

We show that for $\beta-$smooth functions $f$, the $i-$th eigenvalue function:

$$\lambda_i(\theta) = i - \text{th largest eigenvalue of } \nabla_\theta^2 f(\theta) \tag{3}$$

And (an appropriate definition of) the $i-$th eigenvector function:

$$e_i(\theta) = i - \text{th largest normalized eigenvector of } \nabla_\theta^2 f(\theta) \tag{4}$$

are continuous functions of $\theta$.

**Lemma 1.** *If $f$ is $\beta-$smooth' $\lambda_i(\theta)$ is continuous.*

*Proof.* We show the result for the case $i = 1$, the proof for $i \neq 1$ follows the same structure, albeit making use of the more complex variational characterization of the $i-$th eigenvalue / eigenvector pair. Recall the variational formulation of $\lambda_1$:

$$\lambda_1(\theta) = \max_{v \in \mathcal{S}_d(1)} v^\top \nabla_\theta^2 f(\theta) v \tag{5}$$

Let $\theta' = \theta + \Delta_\theta$. It is enough to show that:

$$\lim_{\Delta_\theta \to 0} \lambda_1(\theta') = \lambda_1(\theta)$$

Let $v_1$ be a unit vector achieving the max in Equation 5 and let $v'_1$ be the maximizer for the corresponding variational equation for $\theta'$, then:

$$|\lambda_1(\theta) - v_1^\top \nabla_\theta^2 f(\theta')v_1| = |v_1^\top \left(\nabla_\theta^2 f(\theta) - \nabla_\theta^2 f(\theta')\right)v_1|$$
$$\overset{(i)}{\leq} \beta\|\Delta_\theta\|$$

Inequality $(i)$ follows by $\beta$−smoothness. Similarly:

$$|\lambda_1(\theta') - (v'_1)\nabla_\theta^2 f(\theta)v'_1| \leq \beta\|\Delta_\theta\|$$

Since by definition:
$$\lambda_1(\theta) \geq (v'_1)^\top \nabla_\theta^2 f(\theta)v'_1$$

And:
$$\lambda_1(\theta') \geq v_1^\top \nabla_\theta^2 f(\theta')v_1$$

We conclude that:
$$\lambda_1(\theta') \geq \lambda_1(\theta) - \beta\|\Delta_\theta\|$$

And:
$$\lambda_1(\theta) \geq \lambda_1(\theta') - \beta\|\Delta_\theta\|$$

Consequently:
$$\lambda_1(\theta) + \beta\|\Delta_\theta\| \geq \lambda_1(\theta') \geq \lambda_1(\theta) - \beta\|\Delta_\theta\|$$

The result follows by taking the limit as $\Delta_\theta \to 0$. $\square$

In fact the proof above shows even more:

**Proposition 1.** *If $L$ is $\beta$−smooth and $\theta, \theta' \in \mathbb{R}^d$ then the eigenvalue function is $\beta$-Lipschitz:*
$$|\lambda_i(\theta') - \lambda_i(\theta)| \leq \beta\|\theta - \theta'\|$$

*And therefore:*
$$\|\nabla_\theta \lambda_i(\theta)\| \leq \beta$$

*Proof.* With the exact same sequence of steps as in the proof of Lemma 1, we conclude:
$$\lambda_i(\theta) + \beta\|\Delta_\theta\| \geq \lambda_i(\theta') \geq \lambda_i(\theta) - \beta\|\Delta_\theta\|$$

And:
$$\lambda_i(\theta') + \beta\|\Delta_\theta\| \geq \lambda_i(\theta) \geq \lambda_i(\theta') - \beta\|\Delta_\theta\|$$

The result follows. The gradient bound is an immediate consequence of the Lipschitz property of $\lambda_1(\cdot)$. $\square$

Let's define a canonical $i$−th eigenvector path $\ell : [0, \infty) \to \mathbb{R}$ starting at $\theta$ as follows:

$$\ell(0) = \theta$$

$$\frac{\partial \ell(t)}{\partial t} = \begin{cases} e_i(\theta) & \text{if } \lim_{l \to t}\langle e_i(\theta), \frac{\partial \ell(l)}{\partial l}\ell(l)\rangle > 0 \\ -e_i(\theta) & \text{o.w.} \end{cases}$$

We proceed to show that the curve traced by $\ell$ is continuous.

**Lemma 2.** *Let $\theta$ be such that $\lambda_{i-1}(\theta) - \lambda_i(\theta) = \Delta_{i-1} > 0$ and $\lambda_i(\theta) - \lambda_{i+1}(\theta) = \Delta_i > 0$. Then:*

$$\min(\|e_i(\theta) - e_i(\theta')\|, \|e_i(\theta) + e_i(\theta')\|) \leq \sqrt{\frac{4\beta\|\theta - \theta'\|}{\min(\Delta_i, \Delta_{i-1})}} \tag{6}$$

*For all $\theta'$ such that $\|\theta' - \theta\| \leq \frac{\min(\Delta_{i-1}, \Delta_i)}{4\beta}$*

*Proof.* By Proposition 1, for all $\theta'$ such that $\|\theta' - \theta\| \leq \frac{\min(\Delta_{i-1}, \Delta_i)}{4\beta}$:

$$\max(|\lambda_i(\theta') - \lambda_i(\theta)|, |\lambda_{i-1}(\theta') - \lambda_{i-1}(\theta)|, |\lambda_{i+1}(\theta') - \lambda_{i+1}(\theta)|) \leq \beta\|\theta' - \theta\| \qquad (7)$$

In other words, it follows that:

$$\lambda_{i+1}(\theta), \lambda_{i+1}(\theta') < \lambda_i(\theta), \lambda_i(\theta') < \lambda_{i-1}(\theta), \lambda_{i-1}(\theta')$$

Where this sequence of inequalities implies for example that $\lambda_{i+1}(\theta') < \lambda_i(\theta)$.

Let $H = \nabla_\theta^2 f(\theta)$ and $H' = \nabla_\theta^2 f(\theta')$. Let $\Delta = H - H'$. By $\beta-$smoothness we know that $\|\Delta\| \leq \beta\|\theta - \theta'\|$.

Again for simplicity we restrict ourselves to the case $i = 1$. The argument for $i \neq 1$ uses the same basic ingredients, but takes into account the more complex variational characterization of the $i-$th eigenvector.

W.l.o.g. define $e_1(\theta)$ and $e_1(\theta')$ such that $\langle e_1(\theta), e_1(\theta') \rangle = \alpha \geq 0$. We can write $e_1(\theta') = \alpha e_1(\theta) + \left(\sqrt{1-\alpha^2}\right) v$ with $\|v\|_2 = 1$ and $\langle v, e_1(\theta) \rangle = 0$. Notice that $\|e_1(\theta) - e_1(\theta')\| = \|(1-\alpha)e_1(\theta) - \left(\sqrt{1-\alpha^2}\right) v\| \leq (1-\alpha) + \sqrt{1-\alpha^2}$. We now show $\alpha$ is close to 1. Recall:

$$e_1(\theta) = \arg\max_{v \in \mathbb{S}_d} v^\top H v$$

and

$$e_1(\theta') = \arg\max_{v' \in \mathbb{S}_d} (v')^\top H' v'$$

Equation 7 implies $\lambda_1(\theta) \geq \lambda_2(\theta') + \Delta_1 - \beta\|\theta - \theta'\|$ and $\lambda_1(\theta') \geq \lambda_2(\theta) + \Delta_1 - \beta\|\theta - \theta'\|$. The following inequalities hold:

$$\begin{aligned}
\lambda_1(\theta') &\geq e_1(\theta)^\top H' e_1(\theta) && (8) \\
&= e_1(\theta)^\top [H - \Delta] e_1(\theta) \\
&= e_1(\theta)^\top H e_1(\theta) - e_1(\theta)^\top \Delta e_1(\theta) \\
&= \lambda_1(\theta) - e_1(\theta)^\top \Delta e_1(\theta) \\
&\geq \lambda_1(\theta) - \|\Delta\| \\
&\geq \lambda_1(\theta) - \beta\|\theta - \theta'\| && (9)
\end{aligned}$$

Write $e_1(\theta) = \sum_i \alpha_i e_i(\theta')$ with $\sum_i \alpha_i^2 = 1$. Notice that:

$$e_1(\theta)^\top H' e_1(\theta) = \sum_i \alpha_i^2 \lambda_i(\theta')$$

Since $\lambda_i(\theta') < \lambda_1(\theta) - \frac{3\Delta_1}{4}$ for all $i > 1$:

$$\begin{aligned}
\sum_i \alpha_i^2 \lambda_i(\theta') &\leq \alpha_1^2 \lambda_1(\theta') + \left(\sum_{i=2}^d \alpha_i^2\right) (\lambda_1(\theta) - \Delta_1 + \beta\|\theta - \theta'\|) \\
&\leq \alpha_1^2(\lambda_1(\theta) + \beta\|\theta - \theta'\|) + \left(\sum_{i=2}^d \alpha_i^2\right)(\lambda_1(\theta) - \Delta_1 + \beta\|\theta - \theta'\|) \\
&\leq \lambda_1(\theta) + \beta\|\theta - \theta'\| - \Delta_1(1 - \alpha_1^2) && (10)
\end{aligned}$$

And therefore, combining Equation 9 and 10:

$$\lambda_1(\theta) - \beta\|\theta - \theta'\| \leq e_1(\theta)^\top H' e_1(\theta) \leq \lambda_1(\theta) + \beta\|\theta - \theta'\| - \Delta_1(1 - \alpha_1^2)$$

Therefore:

$$\alpha_1^2 \geq \frac{\Delta_1 - 2\beta\|\theta - \theta'\|}{\Delta_1} = 1 - \frac{2\beta\|\theta - \theta'\|}{\Delta_1}$$

This in turn implies that $\sum_{i=2}^{d} \alpha_i^2 \leq \frac{2\beta\|\theta-\theta'\|}{\Delta_1}$ and that $1-\alpha \leq 1-\alpha^2 \leq \frac{2\beta\|\theta-\theta'\|}{\Delta_1}$. Therefore:

$$\|e_1(\theta) - e_1(\theta')\|^2 = (1-\alpha_1)^2 + \sum_{i=2}^{d} \alpha_2^2$$

$$\leq (1-\alpha_1)^2 + \frac{2\beta\|\theta-\theta'\|}{\Delta_1}$$

$$\leq \frac{4\beta^2\|\theta-\theta'\|^2}{\Delta_1^2} + \frac{2\beta\|\theta-\theta'\|}{\Delta_1}$$

$$\leq \frac{4\beta\|\theta-\theta'\|}{\Delta_1}$$

The result follows.

$\square$

As a direct implication of Lemma 2, we conclude the eigenvector function is continuous.

### D.3 Convergence rates for finding a new eigenvector, eigenvalue pair

Let $\theta' = \theta + \Delta_\theta$, ridge riding minimizes the following loss w.r.t $e$ and $\lambda$ to find a candidate $e'$ and $\lambda'$:

$$L(e, \lambda; \theta') = \|(1/\lambda)\mathcal{H}(\theta')e/\|e\| - e/\|e\|\|^2 \tag{11}$$

Notice that:

$$L(e, \lambda; \theta') = \frac{1}{\lambda^2\|e\|^2}e^\top\mathcal{H}(\theta')^2 e + 1 - 2\frac{1}{\lambda\|e\|^2}e^\top\mathcal{H}(\theta')e$$

Therefore:

$$\nabla_e L(e, \lambda; \theta') = \frac{1}{\lambda^2}\nabla_e\left(\frac{1}{\|e\|^2}e^\top\mathcal{H}(\theta')^2 e\right) - 2\frac{1}{\lambda}\nabla_e\left(\frac{1}{\|e\|^2}e^\top\mathcal{H}(\theta')e\right)$$

$$= \frac{2}{\lambda^2\|e\|}\left(\mathcal{H}^2(\theta') - \tilde{e}^\top\mathcal{H}^2(\theta')\tilde{e}I\right)\tilde{e} - \frac{4}{\lambda\|e\|}\left(\mathcal{H}(\theta') - \tilde{e}^\top\mathcal{H}(\theta')\tilde{e}I\right)\tilde{e}$$

$$= \left(\frac{2}{\lambda^2\|e\|}\mathcal{H}^2(\theta') - \frac{4}{\lambda\|e\|}\mathcal{H}(\theta')\right)\tilde{e} + \left(\frac{4}{\lambda\|e\|}\tilde{e}^\top\mathcal{H}(\theta')\tilde{e}I - \frac{2}{\lambda^2\|e\|}\tilde{e}^\top\mathcal{H}^2(\theta')\tilde{e}I\right)\tilde{e}$$

Where $\tilde{e} = \frac{e}{\|e\|}$.

Now let's compute the following gradient:

$$\nabla_\lambda L(e, \lambda; \theta') = -\frac{2}{\lambda^3\|e\|^2}e^\top\mathcal{H}(\theta')^2 e + \frac{2}{\lambda^2\|e\|^2}e^\top\mathcal{H}(\theta')e = \frac{2}{\lambda^2\|e\|}\left(e^\top\mathcal{H}(\theta')e - \frac{e^\top\mathcal{H}(\theta')^2 e}{\lambda}\right)$$

We consider the following algorithm:

1. Start at $(e, \lambda)$.
2. Take a gradient step $e \to e - \alpha_e \nabla_e L(e, \lambda; \theta')$.
3. Take a gradient step $\lambda \to \lambda - \alpha_\lambda \nabla_\lambda L(e, \lambda; \theta')$.
4. Normalize $e \to \frac{e}{\|e\|}$.

It is easy to see that the update for $e$ takes the form:

$$e \to \underbrace{\left(\left(1 + \alpha_e\left(\frac{2e^\top\mathcal{H}(\theta')^2 e}{\lambda^2} - \frac{4e^\top\mathcal{H}(\theta')e}{\lambda}\right)\right)I + \alpha_e\left(\frac{4\mathcal{H}(\theta')}{\lambda} - \frac{2\mathcal{H}(\theta')^2}{\lambda^2}\right)\right)}_{U}e$$

$$e \to \frac{e}{\|e\|}$$

Where we think of $U$ as an operator acting on the vector $e$. In fact if we consider $T$ consecutive steps of this algorithm, yielding normalized eigenvector candidates $e_0, \cdots, e_T$ and eigenvalue candidates $\lambda_0, \cdots, \lambda_T$, and name the corresponding $U-$operators as $U_1, \cdots, U_T$ it is easy to see that:

$$e_T = \frac{E_T}{\|E_T\|}$$

Where $E_T = \left(\prod_{i=1}^{T} U_T\right) e_0$. In other words, the normalization steps can be obviated as long as we normalize at the very end. This observation will prove useful in the analysis.

Let's assume $L$ is $\beta-$smooth and let's say we are trying to find the $i-$th eigenvalue eigenvector pair for $\theta'$: $(e_i(\theta'), \lambda_i(\theta'))$. Furthermore let's assume we start our optimizaation at the $(e_i(\theta), \lambda_i(\theta))$ pair. Furthermore, assume that $\theta'$ is such that:

$$\|e_i(\theta) - e_i(\theta')\| \leq \min(\frac{\Delta_i}{4}, \frac{\Delta_{i-1}}{4}) \text{ and } \|\lambda_i(\theta) - \lambda_i(\theta')\| \leq \min(\frac{\Delta_i}{4}, \frac{\Delta_{i-1}}{4}) \qquad (12)$$

Where $\lambda_i(\theta) - \lambda_i(\theta) = \Delta_{i-1} > 0$ and $\lambda_i(\theta) - \lambda_{i+1}(\theta) = \Delta_i > 0$. The existence of such $\theta'$ as in 12 can be guaranteed by virtue of Lemmas 1 and 2.

Notice that as long as $\alpha_e = \min(1/4, \Delta_i, \Delta_{i-1})$ is small enough the operator $U$ attains the form:

$$U = AI + \alpha_e \left(\frac{4\mathcal{H}(\theta')}{\lambda} - \frac{2\mathcal{H}(\theta')^2}{\lambda^2}\right)$$

Where $\alpha_e/A$ is small.

Notice that the operator $\left(\frac{4\mathcal{H}(\theta')}{\lambda} - \frac{2\mathcal{H}(\theta')^2}{\lambda^2}\right)$ has the following properties:

1. $\left(\frac{4\mathcal{H}(\theta')}{\lambda} - \frac{2\mathcal{H}(\theta')^2}{\lambda^2}\right)$ has the exact same eigenvectors set $\{e_j(\theta')\}_{j=1}^T$ as $\mathcal{H}(\theta')$.

2. The eigenvalues of $\left(\frac{4\mathcal{H}(\theta')}{\lambda} - \frac{2\mathcal{H}(\theta')^2}{\lambda^2}\right)$ equal $\{\frac{4\lambda_j(\theta')}{\lambda} - \frac{2\lambda_j(\theta'^2}{\lambda^2}\}_{j=1}^d$.

Consequently, if $|\lambda - \lambda_i(\theta')| < \min(\frac{\Delta_i'}{4}, \frac{\Delta_{i-1}'}{4})$ ) we conclude that the maximum eigenvalue of $\left(\frac{4\mathcal{H}(\theta')}{\lambda} - \frac{2\mathcal{H}(\theta')^2}{\lambda^2}\right)$ equals $\frac{4\lambda_i(\theta')}{\lambda} - \frac{2\lambda_i(\theta')^2}{\lambda^2}$ with eigenvector $e_i(\theta')$.

Furthermore, the eigen-gap between the maximum eigenvalue and any other one is lower bounded by $\frac{\min(\Delta_i, \Delta_{i-1})}{2}$. Therefore, after taking a gradient step on $e$, the dot product $\langle e_i(\theta'), e_t \rangle = \gamma_t$ satsifies $\gamma_{t+1}^2 \rightarrow \gamma_t^2 + \gamma_t^2 * (1 - \alpha_e^2 \frac{\min(\Delta_i, \Delta_{i-1}^2)}{4})$

If $|\lambda_t - \lambda_i(\theta')| < \min(\frac{\Delta_i'}{4}, \frac{\Delta_{i-1}'}{4})$ ), and the eigenvalue update satisfied the properties above, then $\lambda_{t+1}$ is closer to $\lambda_i(\theta')$ than $\lambda_t$, thus maintaining the invariance. We conclude that the convergence rate is the rate at which $\gamma_t \rightarrow 1$, which is captured by the following theorem:

**Theorem 3.** *If $L$ is $\beta-$smooth, $\alpha_e = \min(1/4, \Delta_i, \Delta_{i-1})$, and $\|\theta - \theta'\| \leq \frac{\min(1/4, \Delta_i, \Delta_{i-1})}{\beta}$ then*

$$|\langle e_t, e_i(\theta')\rangle| \geq 1 - \left(1 - \frac{\min(1/4, \Delta_i, \Delta_{i-1})}{4}\right)^t$$

### D.4 Staying on the ridge

In this section, we show that under the right assumptions on the step sizes, Ridge Riding stays along a descent direction.

We analyze the following setup. Starting at $\theta$, we move along negative eigenvector $e_i(\theta)$ to $\theta' = \theta - \alpha e_i(\theta)$. Once there we move to $\theta'' = \theta' - \alpha e_i(\theta')$. Let $L : \Theta \rightarrow \mathbb{R}$ be the function we are trying to optimize. We show that:

**Theorem 4.** *Let $L : \Theta \rightarrow \mathbb{R}$ have $\beta-$smooth Hessian, let $\alpha$ be the step size. If at $\theta$ RR satisfies: $\langle \nabla L(\theta), e_i(\theta)\rangle \geq \|\nabla L(\theta)\|\gamma$, and $\alpha \leq \frac{\min(\Delta_i, \Delta_{i-1})\gamma^2}{16\beta}$ then after two steps of RR:*

$$L(\theta'') \leq L(\theta) - \gamma\alpha\|\nabla L(\theta)\|$$

*Proof.* Since $L$ is $\beta-$smooth, the third order derivatives of $L$ are uniformly bounded. Let's write $L(\theta')$ using a Taylor expansion:

$$L(\theta') = L(\theta - \alpha e_i(\theta))$$
$$\overset{(i)}{\leq} L(\theta) + \langle \nabla L(\theta), -\alpha e_i(\theta) \rangle + \frac{1}{2}(-\alpha e_i(\theta)^\top \mathcal{H}(\theta)(-\alpha e_i(\theta)) + c'\alpha^3 \beta$$
$$= L(\theta) - \alpha \langle \nabla L(\theta), e_i(\theta) \rangle + \alpha^2 \frac{\lambda_i(\theta)}{2} + c'\alpha^3 \beta$$

Inequality $(i)$ follows by Hessian smoothness.

Let's expand $L(\theta'')$:

$$L(\theta'') = L(\theta' - \alpha e_i(\theta'))$$
$$\leq L(\theta') - \alpha \langle \nabla L(\theta'), e_i(\theta') \rangle + \frac{\alpha^2}{2}\lambda_i(\theta') + c''\alpha^3 \beta$$
$$\leq L(\theta) - \alpha \langle \nabla L(\theta), e_i(\theta) \rangle - \alpha \langle \nabla L(\theta'), e_i(\theta') \rangle + \frac{\alpha^2 \lambda_i(\theta)}{2} + \frac{\alpha^2 \lambda_i(\theta')}{2} + (c' + c'')\alpha^3 \beta$$

Notice that for any $v \in \mathbb{R}^d$ it follows that $\langle \nabla L(\theta'), v \rangle \leq \langle \nabla L(\theta), v \rangle - \alpha v^\top \nabla^2 L(\theta) e_i(\theta) + c'''\beta \alpha^2 = \langle \nabla L(\theta), v \rangle - \alpha \lambda_i(\theta) v^\top e_i(\theta) + c'''\beta \alpha^2$. Plugging this in the sequence of inequalities above:

$$L(\theta'') \leq L(\theta) - \alpha \langle \nabla L(\theta), e_i(\theta) \rangle - \alpha \langle \nabla L(\theta'), e_i(\theta') \rangle + \frac{\alpha^2 \lambda_i(\theta)}{2} + \frac{\alpha^2 \lambda_i(\theta')}{2} + (c' + c'')\alpha^3 \beta$$
$$\leq L(\theta) - \alpha \langle \nabla L(\theta), e_i(\theta) \rangle - \alpha \langle \nabla L(\theta), e_i(\theta') \rangle + \frac{3\alpha^2 \lambda_i(\theta) + \alpha^2 \lambda_i(\theta')}{2} + (c' + c'' + c''')\alpha^3 \beta$$
$$= L(\theta) - 2\alpha \langle \nabla L(\theta), e_i(\theta) \rangle + \alpha \langle \nabla L(\theta), e_i(\theta) - e_i(\theta') \rangle + \frac{3\alpha^2 \lambda_i(\theta) + \alpha^2 \lambda_i(\theta')}{2} + (c' + c'' + c''')\alpha^3 \beta$$
$$\leq L(\theta) - 2\alpha \langle \nabla L(\theta), e_i(\theta) \rangle + \alpha \langle \nabla L(\theta), e_i(\theta) - e_i(\theta') \rangle + \frac{3\alpha^2 \lambda_i(\theta) + \alpha^2 \lambda_i(\theta')}{2} + (c' + c'' + c''')\alpha^3 \beta$$
$$\overset{(i)}{\leq} L(\theta) - 2\alpha \langle \nabla L(\theta), e_i(\theta) \rangle + \alpha \|\nabla L(\theta)\| \sqrt{\frac{4\beta \|\theta - \theta'\|}{\min(\Delta_i, \Delta_{i-1})}} + \frac{3\alpha^2 \lambda_i(\theta) + \alpha^2 \lambda_i(\theta')}{2} + (c' + c'' + c''')\alpha^3 \beta$$
(13)

Where inequality $(i)$ follows from Cauchy-Schwarz and Lemma 2 since:

$$\|e_i(\theta) - e_i(\theta')\| \leq \sqrt{\frac{4\beta \|\theta - \theta'\|}{\min(\Delta_i, \Delta_{i-1})}} = \sqrt{\frac{4\beta \alpha}{\min(\Delta_i, \Delta_{i-1})}}$$

Recall that by assumption $\langle \nabla L(\theta), e_i(\theta) \rangle \geq \|\nabla L(\theta)\|\gamma$ and $\gamma \in (0,1)$ and that $\alpha \leq \frac{\min(\Delta_i, \Delta_{i-1})\gamma^2}{16\beta}$. Applying this to inequality 13:

$$L(\theta'') \leq L(\theta) - 2\alpha \langle \nabla L(\theta), e_i(\theta) \rangle + \alpha \|\nabla L(\theta)\| \sqrt{\frac{4\beta \alpha}{\min(\Delta_i, \Delta_{i-1})}} + \frac{3\alpha^2 \lambda_i(\theta) + \alpha^2 \lambda_i(\theta')}{2} + (c' + c'' + c''')\alpha^3 \beta$$
$$\leq L(\theta) - 2\alpha \langle \nabla L(\theta), e_i(\theta) \rangle + \frac{\gamma \alpha}{2}\|\nabla L(\theta)\| + \frac{3\alpha^2 \lambda_i(\theta) + \alpha^2 \lambda_i(\theta')}{2} + (c' + c'' + c''')\alpha^3 \beta$$
$$\leq L(\theta) - \gamma \alpha \|\nabla L(\theta)\| + \frac{3\alpha^2 \lambda_i(\theta) + \alpha^2 \lambda_i(\theta')}{2} + (c' + c'' + c''')\alpha^3 \beta$$
$$\leq L(\theta) - \gamma \alpha \|\nabla L(\theta)\|$$

The last inequality follows because term $\frac{3\alpha^2 \lambda_i(\theta) + \alpha^2 \lambda_i(\theta')}{2} \leq 0$ and of order less than the third degree terms at the end.

$\square$

## D.5 Behavior of RR near a saddle point

The discussion in this section is intended to be informal and has deliberately been written in this way. First, let $\theta_0$ be a saddle point of $\mathcal{L}(\theta)$, and consider the steps of RR $\theta_1, ..., \theta_t, \cdots$. Let $H$ be the Hessian of $\mathcal{L}$ at $\theta_0$. We will start by using the first-order Taylor expansion, $\nabla_\theta \mathcal{L}(\theta_t) = \mathcal{H}(\theta_{t-1})(\theta_t - \theta_{t-1}) + o(\epsilon^2)$ ignoring the error term to approximate the gradient close to $\theta_{t-1}$.

We will see that $\nabla_\theta \mathcal{L}(\theta_t) = \alpha \sum_{l=0}^{t-1} \lambda_i(\theta_l) e_i(\theta_l) + o(t\epsilon^2)$ for all $t$. We proceed by induction. Notice that for $t = 1$, this is true since $\nabla_\theta \mathcal{L}(\theta_1) = \mathcal{H}(\theta_0)(\theta_1 - \theta_0) + o(\epsilon^2) = \alpha \lambda_i(\theta_0)) e_i(\theta_0) + o(\epsilon^2)$ for some lower order error term $\epsilon$.

Now suppose that for some $t \geq 1$ we have $\nabla_\theta \mathcal{L}(\theta_t) = \alpha \sum_{l=0}^{t-1} \lambda_i(\theta_l) e_i(\theta_l) + o(t\epsilon^2)$, this holds for $t = 0$. By a simple Taylor expansion around $\theta_t$:

$$
\begin{aligned}
\nabla_\theta \mathcal{L}(\theta_{t+1}) &= \nabla_\theta \mathcal{L}(\theta_t) + \mathcal{H}(\theta_t)(\theta_{t+1} - \theta_t) + o(\epsilon^2) \\
&\overset{(i)}{=} \alpha \sum_{l=0}^{t-1} \lambda_i(\theta_l) e_i(\theta_l) + o(t\epsilon^2) + \mathcal{H}(\theta_t)(\theta_{t+1} - \theta_t) + o(\epsilon^2) \\
&= \alpha \sum_{l=0}^{t} \lambda_i(\theta_l) e_i(\theta_l) + o((t+1)\epsilon^2)
\end{aligned}
$$

Equality $(i)$ follows from the inductive assumption. The last inequality follows because by definition $\mathcal{H}(\theta_t)(\theta_{t+1} - \theta_t) = \alpha \lambda_i(\theta_t) e_i(\theta_t)$. The result follows.

## D.6 Symmetries lead to repeated eigenvalues

Let $\mathcal{L} : \mathbb{R}^d \to \mathbb{R}^d$ be a twice-differentiable loss function and write $[n] = \{1, \ldots, n\}$ for $n \in \mathbb{N}$. For any permutation $\phi \in S_d$ (the symmetric group on $d$ elements), consider the group action

$$
\phi(\theta_1, \ldots, \theta_d) = (\theta_{\phi(1)}, \ldots, \theta_{\phi(d)})
$$

and abuse notation by also writing $\phi : \mathbb{R}^d \to \mathbb{R}^d$ for the corresponding linear map. For any $N, m \in \mathbb{N}$, define the group of permutations

$$
\Phi_N^m = \left\{ \prod_{i=1}^{m} (i, i + mk) \mid k \in [N-1] \right\}^2.
$$

Now assume there are $N$ non-overlapping sets

$$
\{\theta_{k_i^1}, \ldots, \theta_{k_i^m}\}
$$

of $m$ parameters each, with $i \in [N]$, which we can reindex (by reordering parameters) to

$$
\{\theta_{1+m(i-1)}, \ldots, \theta_{mi}\}
$$

for convenience. Assume the loss function is invariant under all permutations of these $N$ sets, namely, $\mathcal{L} \circ \phi = \mathcal{L}$ for all $\phi \in \Phi_N^m$. Our main result is that such parameter symmetries reduce the number of distinct Hessian eigenvalues, cutting down the number of directions to explore by ridge riding.

**Theorem 5.** *Assume that for some $N, m$ we have $\mathcal{L} \circ \phi = \mathcal{L}$ and $\phi(\theta) = \theta$ for all $\phi \in \Phi_N^m$. Then $\nabla^2 \mathcal{L}(\theta)$ has at most $d - m(N-2)$ distinct eigenvalues.*

We first simplify notation and prove a few lemmata.

**Definition 2.** *Write $\Phi = \Phi_N^m$ and $H = \nabla^2 \mathcal{L}(\theta)$. We define an eigenvector $v$ of $H$ to be* trivial *if $\phi(v) = v$ for all $\phi \in \Phi$. We call an eigenvalue* trivial *if all corresponding eigenvectors are trivial.*

**Lemma 3.** *Assume $\mathcal{L} \circ \phi = \mathcal{L}$ for some $\phi \in \Phi$ and $\phi(\theta) = \theta$. If $(v, \lambda)$ is an eigenpair of $H$ then so is $(\phi(v), \lambda)$.*

*Proof.* First notice that $D\phi$ (also written $\nabla\phi$) is constant by linearity of $\phi$, and orthogonal since

$$(D\phi^T D\phi)_{ij} = \sum_k D\phi_{ki} D\phi_{kj} = \sum_k \delta_{\phi(k)i}\delta_{\phi(k)j} = \delta_{ij}\sum_k \delta_{\phi(k)i} = \delta_{ij} = I_{ij}\,.$$

Now applying the chain rule to $\mathcal{L} = \mathcal{L}\circ\phi$ we have

$$D\mathcal{L} = D\mathcal{L}|_\phi \circ D\phi$$

and applying the product rule and chain rule again,

$$D^2\mathcal{L} = D(D\mathcal{L}|_\phi \circ D\phi) = D\phi^T D^2\mathcal{L}|_\phi D\phi + 0$$

since $D^2\phi = 0$. If $\phi(\theta) = \theta$ then we obtain

$$H = D\phi^T H D\phi\,, \quad \text{or equivalently,} \quad H = D\phi H D\phi^T$$

by orthogonality of $D\phi$. Now notice that $\phi$ acts linearly as a matrix-vector product

$$\phi(v) = D\phi \cdot v\,,$$

so any eigenpair $(v, \lambda)$ of $H$ must induce

$$H\phi(v) = (D\phi H D\phi^T)(D\phi v) = D\phi H v = D\phi \lambda v = \lambda D\phi v = \lambda\phi(v)$$

as required. $\quad\square$

**Lemma 4.** *Assume $v$ is a non-trivial eigenvector of $H$ with eigenvalue $\lambda$. Then $\lambda$ has multiplicity at least $N-1$.*

*Proof.* Since $v$ is non-trivial, there exists $\phi \in \Phi$ such that $\phi(v)_i \neq v_i$ for some $i \in [d]$. Without loss of generality, by reordering the parameters, assume $i = 1$. Since $\phi = \prod_{i=1}^m (i, i+mk)$ for some $k \in [N-1]$, we can set $k$ to $N-1$ after reindexing of the $N$ sets. Now $u = v - \phi(v)$ is an eigenvector of $H$ with $u_1 \neq 0$ and zeros everywhere except the first and last $m$ entries, since $k = N-1$ implies that $\phi$ keeps other entries fixed. We claim that

$$\left\{\prod_{i=1}^m (i, i+mk)u\right\}_{k=0}^{N-2}$$

are $N-1$ linearly independent vectors. Assume there are real numbers $a_0, \ldots, a_{N-2}$ such that

$$\sum_{k=0}^{N-2} a_k \prod_{i=1}^m (i, i+mk)u = 0\,.$$

In particular, noticing that $u_{1+mj} = 0$ for all $1 \leq j \leq N-2$ and considering the $(1+mj)$th entry for each such $j$ yields

$$0 = \sum_{k=0}^{N-2} a_k \left(\prod_{i=1}^m (i, i+mk)u\right)_{1+mj} = \sum_{k=0}^{N-2} a_k \delta_{jk} u_1 = a_j u_1\,.$$

This implies $a_j = 1$ for all $1 \leq j \leq N-2$. Finally we are left with

$$0 = a_0 \prod_{i=1}^m (i, i)u = a_0 u$$

which implies $a_0 = 0$, so the vectors are linearly independent. By the previous lemma, each vector is an eigenvector with eigenvalue $\lambda$, so the eigenspace has dimension at least $N-1$ as required. $\quad\square$

**Lemma 5.** *There are at most $d - m(N-1)$ linearly independent trivial eigenvectors.*

*Proof.* Assume $v$ is a trivial eigenvector, namely, $\phi(v) = v$ for all $\phi \in \Phi$. Then $v_i = v_{i+mk}$ for all $1 \leq i \leq m$ and $1 \leq k \leq N-1$, so $v$ is fully determined by its first $m$ entries $v_1, \ldots, v_m$ and its last $d - mN$ entries $v_{mN+1}, \ldots, v_d$. This implies that trivial eigenvectors have at most $m + d - mN = d - m(N-1)$ degrees of freedom, so there can be at most $d - m(N-1)$ linearly independent such vectors. $\quad\square$

The theorem now follows easily.

*Proof.* Let $k$ and $l$ respectively be the number of distinct trivial and non-trivial eigenvalues. Eigenvectors with distinct eigenvalues are linearly independent, so $k \leq d - m(N-1)$ by the previous lemma. Now assuming for contradiction that $k + l > d - m(N-2)$ implies

$$d - m(N-2) < k + l \leq d - m(N-1) + l \quad \Longrightarrow \quad l > m.$$

On the other hand, each non-trivial eigenvalue has multiplicity at least $N-1$, giving $k + l(N-1) \leq d$ linearly independent eigenvectors. We obtain the contradiction

$$d \geq k + l(N-1) = k + l + l(N-2) > d - m(N-2) + l(N-2) > d$$

and conclude that $k + l \leq d - m(N-2)$, as required. $\qquad\square$

## D.7  Maximally Invariant Saddle

In this section we show that for the case of tabular RL problems, the Maximally Invariant Saddle (MIS) corresponds to the parameter achieving the optimal reward and having the largest entropy.

We consider $\theta \in \mathbb{R}^{|S| \times |A|}$ the parametrization of a policy $\pi_\theta$ over an MDP with states $S$ and actions $A$. We assume $\theta = \{\theta_s\}_{s \in S}$ with $\theta_s \in \mathbb{R}^{|A|}$ and (for simplicity) satisfying[3] $\sum_{a \in A} \theta_{s,a} = 1$ and $\theta_{s,a} \geq 0$.

Let $\Phi$ denote the set of symmetries over parameter space. In other words, $\phi \in \Phi$ if $\phi$ is a permutation over $|S| \times |A|$ and for all $\theta$ a valid policy parametrization, we have that $J(\theta) = J(\phi(\theta))$ such that $\phi(\theta) = \{\phi(\theta_s)\}_{s \in S}$ acting per state.

We also assume the MDP is episodic in its state space, meaning the MDP has a horizon length of $H$ and each state $s \in S$ is indexed by a horizon position $h$. No state is visited twice during an episode.

We show the following theorem:

**Theorem 6.** *Let $\Theta_b$ be the set of parameters that induce policies satisfying $J(\theta) = b$ for all $\theta \in \Theta_b$. Let $\theta^* \in \Theta_b$ be the parameter satisfying $\theta^* = \arg\max_{\theta \in \Theta_b} \sum_s H(\pi_\theta(\mathbf{a}|s))$. Then for all $\phi \in \Phi$ it follows that $\phi(\theta^*) = \theta_*$.*

*Proof.* Let $\theta \in \Theta_b$ and let's assume there is a $\theta' \in \Theta_b$ such that $\phi(\theta') \neq \theta$. We will show there must exist $\theta'' \in \Theta_b$ such that $\sum_s H(\pi_{\theta''}(\mathbf{a}|s)) > \max\left(\sum_s H(\pi_\theta(\mathbf{a}|s)), \sum_s H(\pi_{\theta'}(\mathbf{a}|s))\right)$.

Let $s$ be a state such that $\theta_s \neq \theta'_s$ and having maximal horizon position index $h$. In this case, all states $s'$ with a horizon index larger than $h$ satisfy $\theta_{s'} = \phi(\theta_{s'})$. Therefore for any $s'$ having index $h+1$ (if any) it follows that the value function $V_\theta(s') = V_{\theta'}(s')$. Since the symmetries hold over any policy and specifically for delta policies, it must be the case that at state $s$ and for any $a, a' \in A$ such that there is a $\phi' \in \Phi$ with (abusing notation) $\phi'(s,a) \to s, a'$ it must hold that $Q_\theta(s,a) = Q_\theta(s,a')$. Therefore the whole orbit of $a$ under $\phi'$ for any $\phi' \in A$ has the same $Q$ value under $\theta$ and $\theta'$. Since the entropy is maximized when all the probabilities of these actions are the same, this implies that if $\theta$ does not correspond to a policy acting uniformly over the orbit of $a$ at state $s$ we can increase its entropy by turning it into a policy that acts uniformly over it. Applying this argument recursively down the different layers of the episodic MDP implies that for any $a \in A$, the maximum entropy $\theta \in \Theta_b$ assigns a uniform probability over all the actions on $a$'s orbit. It is now easy to see that such a policy must satisfy $\phi(\theta) = \theta$ for all $\phi \in \Phi$.

$\qquad\square$

We now show a result relating the entropy regularized gradient-norm objective:

$$\arg\min_\theta |\nabla_\theta J(\theta)| - \lambda H(\pi_\theta(\mathbf{a})), \lambda > 0$$

In this discussion we will consider a softmax parametrization for the policies. Let's start with the following lemma:

**Lemma 6.** *Let* $p_i(\theta) = \frac{\exp(\theta_i)}{\sum_j \exp(\theta_j))}$ *parametrize a policy over $K$ reward values $\{r_i\}_{i=1}^K$. The value function's gradient satisfies:*

$$\left(\nabla \sum_{j=1}^K p_j(\theta)\right)_i = p_\theta(i)(r_i - \bar{r})$$

*Where $\bar{r} = \sum_{j=1}^K p_i(\theta) r_i$.*

*Proof.* Let $Z(\theta) = \sum_{j=1}^K \exp(\theta_j)$. The following equalities hold:

$$\left(\nabla \sum_{j=1}^K p_j(\theta)\right)_i = \frac{Z(\theta)\exp(\theta_i)r_i - \exp^2(\theta_i)r_i}{Z^2(\theta)} + \sum_{j\neq i} \frac{-\exp(\theta_j)r_j\exp(\theta_i)}{Z^2(\theta)}$$

$$= \frac{Z(\theta)\exp(\theta_i)r_i}{Z^2(\theta)} - \sum_j \exp(\theta_i)\frac{\exp(\theta_j)}{Z^2(\theta)}$$

$$= \frac{\exp(\theta_i)r_i}{Z(\theta)} - \frac{\exp(\theta_i)}{Z(\theta)}\left(\sum_j \frac{\exp(\theta_j)r_j}{Z(\theta)}\right)$$

$$= p_i(\theta)\left(r_i - \bar{r}\right).$$

The result follows. $\qquad\square$

We again consider an episodic MDP with horizon length of $H$ and such that each state $s \in S$ is indexed by a horizon position $h$. No state is visited twice during an episode. Recall the set of symmetries is defined as $\phi \in \Phi$ if for any policy $\pi : \mathcal{S} \to \Delta_A$, with $Q-$function $Q_\pi : \mathcal{S} \times \mathcal{A} \to \mathbb{R}$, it follows that:

$$Q_\pi(\phi(s), \phi(a)) = Q_\pi(s, a). \tag{14}$$

We abuse notation and for any policy parameter $\theta \in \mathbb{R}^{|\mathcal{S}| \times |\mathcal{A}|}$ we denote the parameter vector resulting of the action of a permutation on $\phi$ on the indices of a parameter vector $\theta$ by $\phi(\theta)$.

We show the following theorem:

**Theorem 7.** *Let $\Theta_b$ be the set of parameters that induce policies satisfying $\|\nabla J(\theta)\| = b$ for all $\theta \in \Theta_b$. Let $\theta^* \in \Theta_b$ be a parameter satisfying $\theta^* = \arg\max_{\theta \in \Theta_b} \sum_s H(\pi_\theta(\mathbf{a}|s))$ (there could be multiple optima). Then for all $\phi \in \Phi$ it follows that $\phi(\theta^*) = \theta_*$.*

*Proof.* Let $\theta \in \Theta_b$ and let's assume there is a $\phi \in \Phi$ such that $\theta' = \phi(\theta) \neq \theta$.

We will show there must exist $\theta'' \in \Theta_b$ such that $\sum_s H(\pi_{\theta''}(\mathbf{a}|s)) > \max\left(\sum_s H(\pi_\theta(\mathbf{a}|s)), \sum_s H(\pi_{\theta'}(\mathbf{a}|s))\right)$.

Since we are assuming a softmax parametrization and $\phi$ is a symmetry of the MDP, it must hold that for any two states $s$ and $s'$ with (abusing notation) $s' = \phi(s)$:

$$E_{a\sim\pi_\theta}[Q_{\pi(\theta)}(s,a)] = E_{a\sim\pi_{\phi(\theta)}}[Q_{\pi(\phi(\theta))}(s,a)]$$

We conclude that the gradient norm $\|\nabla J(\phi(\theta))\|$ must equal that of $\|\nabla J(\theta)\|$. This implies that if $\phi(s) \neq \phi(s')$ and wlog $H(\pi_\theta(\mathbf{a}|s)) > H(\pi_\theta(\mathbf{a}|\phi(s)))$, then we can achieve the same gradient norm but larger entropy by substituting $\theta_{\phi(s)}$ with $\theta_s$. Where $\theta_s$ denotes the $|\mathcal{A}|$-dimensional vector of the policy parametrization for state $s$. The gradient norm would be preserved and the total entropy of the resulting policy would be larger of that achieved by $\theta$ and $\theta'$. This finalizes the proof.

$\qquad\square$

## Footnotes

[2] For $m = 1$ we have $\Phi_N^1 = \{(1, 2), \ldots, (1, N)\}$, which together generate all $N!$ permutations on $N$ elements. For larger $m$, $\Phi_N^m$ also generates $N!$ permutations, but the $m$ elements within each set are tied to each other. For instance, $\Phi_3^2 = \{(1,3)(2,4), (1,5)(2,6)\}$ which together generate $\{(1), (1,3)(2,4), (1,5)(2,6), (3,5)(4,6), (1,3,5)(2,4,6), (1,5,3)(2,6,4)\}$.

[3]A similar argument follows for a softmax parametrization.