[Reviews · NeurIPS 2020]

Review 1

Summary and Contributions: Authors have introduced Ridge Riding, a novel method for finding specific types of solutions by following ridges - the eigenvectors of the Hessian. Experimental results show that the proposed methods can avoid some obvious minima and explore diverse solutions.

Strengths: As far as I understand, there has been no systematic efforts exploring optimization surface systematically and searching for diverse (unique) solutions. Unlike existing methods which rely on population search or modified objective functions, authors propose a way to explore the surface by selecting different directions formed by the Hessian.

Weaknesses: The method has not been well defined yet. For example, authors are stating that the proposed method starts from a saddle point and follows eigenvectors with negative eigenvalue, but the illustrative example in Figure 1 starts from local maxima. Thus, applying Algorithm 1 would be problematic in this case. Experimental results do show improved diversity but the performance is not satisfactory yet.

Correctness: Theoretically, there is no guarantee that following eigenvectors of Hessian leads to a solution. I agree that there can be an effective strategy utilizing eigenvectors of Hessian, but simply following eigenvalues of Hessian from a saddle point doesn't sound effective enough. The example optimization paths in Figure 3 are good examples. Experiments are well designed and show the potential of the proposed idea.

Clarity: The paper is well written but I feel like the proposed method has not been clearly defined yet, even in authors' minds. line 156: step step -> step

Relation to Prior Work: Yes.

Reproducibility: No

Additional Feedback: Here are some thoughts for more stable method. It is not clear eigenvectors of the Hessian lead to different or diverse solutions, but it is highly likely that avoiding the eigenvector direction of the biggest eigenvalue lets us avoid the closest local minima. How about avoiding eigenvector directions rather than selecting them, until the smallest (and negative) eigenvalues become big enough? Avoiding the smallest eigenvalue and doing steepest descent can still be regarded as 'ridge-riding' in my opinion. The solution can branch at all the points the first eigenvalue becomes as big as the second (and symmetry can also be handled in the same way as proposed in the paper). It is easier to estimate n smallest eigenvalues and removing them from descent direction than to estimate the direction of n+1th eigenvalue and follow it - I believe this approach will improve numerical stability of the solution, too.


Review 2

Summary and Contributions: This paper proposes Ridge Riding (RR) as an optimization approach for machine learning tasks. Unlike gradient descent methods which update the parameters in the “steepest” direction, the proposed RR update parameters along Hessian eigen-directions. The authors expect RR can provide better local minima (in the sense of generalization capability) than gradient descent.

Strengths: The figures presented are very helpful in understanding the main idea of the paper.

Weaknesses: The computational cost of the proposed algorithm is extremely high. >> Let’s say the # of parameters = m, the dimension of Hessian matrix is m*m. A computation of the Hessian spectrum should be O(m^3). Note that for large machine learning models, such as deep neural networks, m is typically over millions. This makes the computation almostly prohibitive. >>The algorithm even requires a spectral computation of the Hessian matrix at EACH timestamp. this should be unrealistic. >>The algorithm keeps track of all the replicas. In the worst case, the number of replicas should be as much as the dimension of Hessian, i.e., number of parameters. This is huge. >>A complexity analysis (both temporal and spatial) of the algorithm is expected. Convergence of the algorithm is not quite guaranteed. Although Thm 1 is about the convergence, the assumption of Thm 1 is not well supported. Thm 1 assumes that (line 187) the projection of each Hessian eigenvector e_i onto the gradient is uniformly greater than a positive value. As the e_i’s and the gradient are changing from time to time, and there are up to m e_i’s in total (m is the # of parameter of the model, which is often more than millions), I doubt if the assumption is not satisfied in practice. The motivation of the paper is to provide better local minima or solutions than gradent descent methods. However, the paper does not justify that these minima provided by RR are better than those generated by gradient descent.

Correctness: As I mentioned in the weakness part, I have doubt that the assumption of Thm 1 may not hold.

Clarity: There are quite a few things which are not clearly delivered. About Algorithm 1: >>Submodules of the algorithm are not explicitly presented. It is not clear that how the the submodules “ChooseFromArchive”, “UpdateRidge” and “GetRidge” are exactly implemented. Although there is description of the intuitions, it is far from a real algorithm. >>What is the output of the algorithm? does it output all the replicas, or a selected one? What is the standard and procedure to select the best parameter setting among all the replicas? About symmetry and equivalence: The paper discussed in Section 3 about the symmetry of loss function and equivalence of parameters. However, it seems totally disappeared when discussing the proposed method. I can not see how the symmetry and equivalence relate to the algorithm.

Relation to Prior Work: Related works are properly mentioned.

Reproducibility: No

Additional Feedback:


Review 3

Summary and Contributions: This is a very interesting work where the authors address the problem of gradient descent approaches converging to arbitrary local minima. They propose an approach where instead of traversing the gradient the optimizer will traverse the ridges along the error surface, towards whichever direction there is negative curvature. Such ridges are obtained via computing the eigenvectors of the Hessian. By exploring along the eigenvectors of the Hessian with highest Eigenvalues it is expected that the most useful and the most diverse solutions will be identified. They term this as ridge riding. They propose both exact and approximate approaches and evaluate the effectiveness of the same on varied kinds of tasks including optimal exploration for diversity in RL, Out of distribution extrapolation etc.

Strengths: 1. The problem itself involves an extremely wide interest group across all areas of ML/AI. The outcome and the insights from this work can be applied to any problem in AI where non-convex optimization is involved. 2. Though the intuition behind the approach is is well known and simple to understand, yet the way the authors adopted the same to find ridges on the loss surface and traversing that instead of looking at the gradient is very clever. 3. The concept of branching among multiple possible ridges using the top eigenvalues recursively while updating the parameters which in turn updates the ridges ... making it a discrete search spaces is a very nice way of transforming the problem (it shows a new connection to the branch and bound technique for optimization ... just in a Hessian space) 4. The presentation and writing of the paper is well done. For instance the authors highlight some important insights into both the problem and the solution approach. For instance Theorem 1, provides a very important theoretical result that is essential towards realizing why the approach might work. 5. The approximate version of the RR algorithm is also a a very nice addition, given that for larger sample spaces eigen decomposition of Hessian computation can be prohibitive and thus updates using Hessian-eigenvector products is a very nice touch. 6. Finally evaluation on 3 important tasks in the paradigm of machine/reinforcement learning solidifies the claim made in the work. 7. Also the Figure 1 plot is very enlightening, if a reader looks at the plot both the problem and the solution will be clear. Very nice work.

Weaknesses: No major concerns, few minor ones such as, 1. In lines 179-182, the authors outline what are the ways in which search order can be determined for selection in ChooseFromArchive. However they did not finally mention what was done in their case and why? 2. Many pre-print archives are cited. Why is that a thing? There is no verifiable validity for the same if it was not peer-reviewed.

Correctness: Yes, claims and results appear correct.

Clarity: Yes the paper is well written

Relation to Prior Work: Yes relevant prior work is discussed. In fact this is one of those works that have adequate related work discussions.

Reproducibility: Yes

Additional Feedback: Just address the concerns raised in the weakness box.


Review 4

Summary and Contributions: Paper describes a method to find diverse solutions of a (neural network) optimization problem by following eigenvectors of the Hessian. A pool of working solutions is maintained; for one of these eigenvectors corresponding to most negative eigenvalues are obtained with a power method; a particular eigenvector (identified by the rank order of its eigenvalue) is then followed. Experiments suggest that doing this yields useful solutions.

Strengths: Experiments on zero-shot coordination involve a clever trick; keeping track of the "fingerprint" (indices of eigenvectors chosen at each branch point in the past) identifies compatible policies for the agents. This removes an inconvenient symmetry from the problem, and is a real strength. There is good experimental evidence that solutions for reinforcement learning are diverse.

Weaknesses: The main difficulty is that it isn't clear that the tremendous expense in finding the relevant eigenvectors is justified in general. The procedure described doesn't necessarily produce diverse solutions. I was unable to produce a theorem either way, but simple numerical experiments on random cubic polynomial cost functions in 2D suggest that sometimes ridge riding solutions diverge, and sometimes they converge. There isn't any math in the paper about this. Given this, one should ask: how would a less involved procedure for producing different descent directions behave? (for example, choose a direction at random, and flip its sign if it isn't a descent direction). One could reasonably expect that the diversity of solutions produced results from having multiple searches using distinct descent directions, rather than from the identity of the directions. Authors could fairly argue that this approach would not have the fingerprint tracking trick, but that doesn't really deal with the concern that there might be an easier way of getting distinct descent directions that does. For example, given the assumption that the eigenvectors of the Hessian don't rotate "too fast" as one moves (eg ll180 et seq), you could recover these eigenvectors very seldom, and maintain some identity based on projections of descent directions on these eigenvectors (which don't change too fast, 190-191).

Correctness: I believe paper to be correct

Clarity: Paper is clear

Relation to Prior Work: Yes

Reproducibility: Yes

Additional Feedback: ---- added post discussion process ----- I stand by review above, after discussion with other referees.

[Author Response · NeurIPS 2020]

We would firstly like to thank the reviewers for their time. We are particularly excited to see that even the reviewers were inspired to propose a variety of extensions and improvements in their feedback, which shows the interest this paper may generate at NeurIPS. In addition, we are pleased to see our algorithm is also capable of producing diverse reviews. Given the green field nature of this project, we have some additional insights to provide since the submission.

**New RL experiments** We extended our RL experiments to the approximate setting where gradients are computed using samples. We use an Actor-Critic algorithm with a tabular policy, and compute Hessians using the DiCE operator (Foerster, 2018). The results show that RR works not only in the exact setting but also when gradients are approximate. We believe this is a key step towards eventually using RR for deep RL. We believe these results improve the potential scalability of our approach, and will be included in the CRC. Below we address individual comments in more detail:

Figure 1: % solutions found per algorithm, by tree depth, each is randomly generated 10 times to produce error estimates shown.

**R1**: **Saddle vs local maximum** When we use the word 'saddle' we mean a stationary point with at least one negative curvature direction, which includes maxima. **method ..not clearly defined .. even in authors' minds.** We provide a method section, multiple instances of pseudo code and implementations that closely match the pseudo code.

**R2**: **computation of the Hessian spectrum should be** $O(m^3)$**..requires a spectral computation EACH timestep..complexity analysis** We addressed this in the section labeled *Approximate Ridge Riding* There, we formulated a version of RR that only requires Hessian-Vector products, which can be computed with $O(m)$ complexity in modern auto-diff frameworks. As explained, we also use an iterative process for updating the EV at each timestep rather than recomputing it. The overall complexity will depend on the search method being used, but each solution will required $O(m) * O(N)$ compute, where $N$ is the number of update steps. The memory is also dependent on the search method, but will be $O(m)$ for depth-first. **Theorem 1 assumption.** One can use Hessian-Vector products to compute $\langle \nabla L(\theta), e_i(\theta) \rangle$ and estimate $\gamma$ at any $\theta$. This can be used to set the learning rate. **minima not better than SGD.** We show in both the zero-shot coordination problem and the out-of-distribution that RR can obtain better solutions than SGD. Clearly though, in general this will be problem specific and all we can do is provide a method that can obtain more diverse solutions. **cannot see how symmetry and equivalence relate to algorithm.** This is exploited and described in the zero-shot coordination setting and in the out-of-distribution part. In general symmetries can be used to make the search more efficient by only exploring one EV from each equivalence group.

**R3**: Many thanks for the encouraging, insightful and positive review! **ChooseFromArchive** The best way to search is problem dependent, so we specify ChooseFromArchive separately in the method sections for each of our applications and extensions of RR. **Many pre-print cited** We apologize. That there are 18 papers in the paper which look like 'pre-prints' is an artifact of us using the default Google Scholar bibtex. Out for these 18, 3 are journal papers, 9 are top-tier conference papers, and only 6 are actual pre-prints. Of those, 4 are from 2020, one is the Tensorflow paper with >5k citations, and the last is the well-known and relevant MAP-elite algorithm. We will correct the citations in the Camera Ready Copy (CRC).

**R7**: Thank you for your interesting comments. We are glad you appreciate the novelty of our work in the ZS setting and note the strength of the diversity in RL experiments. Regarding **RR not working "in generality"**, we feel this is a high bar for any algorithm. While we show that our algorithm is applicable and useful in a wide array of diverse settings, we clearly can't promise that it will find all solutions in all settings (there is probably an impossibility result somewhere to be written down here). **ridges vs. random directions** We tried to address this in Fig 2 (in the paper) with a baseline that follows random (unit) vectors instead of EVs (*Rand. Ridge*). To evaluate your proposal, we include two additional baselines: (1) following ridges, but not updating them (*Fixed-EV*). (2) following random unit vectors with *positive ascent direction* (*Rand-Ridge+*),

Figure 2: Ablations on ridge riding algorithm. Left: We perform a hyperparameter search for all methods on MNIST and show best performance found. Right: The same four methods for diversity in RL, with tree depth of 12, 5 seeds. Legend applies to both plots.

and compare vs. Ridge Riding. We ran these ablations both on the RL experiment (exact RR) and on MNIST, where we used a fixed budget hyperparameter search for approximate RR and the ablations. As shown in Fig 2, RR on MNIST significantly outperforms all ablations. Fixed-EVs obtain a top accuracy of a linear classifier ( 92%), compared to the 98% obtained by RR. In contrast, none of the suggestions using random directions exceed 30%. In the low dimensional RL example, Fixed-EVs obtains competitive performance. This clearly illustrates the importance of following EVs rather than random directions. We furthermore observe all differences to be more pronounced in MNIST, which is intuitive since random search is known to scale poorly to high dimensional problems. We expect this effect to be even more pronounced as Approximate RR is applied to harder and higher dimensional tasks in the future. We once again thank the reviewers for these suggestions and will update the paper to include all plots shown here.

[Meta-Review · NeurIPS 2020]

This paper proposes Ridge Regression, a novel method for exploring local minima using eigenvectors of Hessian. The novelty of the method was appreciated and it was also noted that the method can connect to classic methods such as Branch and Bound technique. There was also appreciation of the empirical results. However, the lack of mathematical insights related to diversity of solutions, --a key claim of the paper-- was missed. Even if there was no analysis a comparison with the random orthonormal set of descent directions in producing diverse directions would have been illustrative to understand the power of the method. The overall consensus is that the paper while having excellent novelty lacks mathematical rigour and thus did not have any strong advocate.